# INFORMATION-THEORETIC CRITERIA FOR KNOWLEDGE DISTILLATION IN MULTIMODAL LEARNING

## ABSTRACT

The rapid increase in multimodal data availability has sparked significant interest in cross-modal knowledge distillation (KD) techniques, where richer "teacher" modalities transfer information to weaker "student" modalities during model training to improve performance. However, despite successes across various applications, cross-modal KD does not always result in improved outcomes, primarily due to a limited theoretical understanding that could inform practice. To address this gap, we introduce the Cross-modal Complementarity Hypothesis (CCH): we propose that cross-modal KD is effective when the mutual information between teacher and student representations exceeds the mutual information between the student representation and the labels. We theoretically validate the CCH in a joint Gaussian model and further confirm it empirically across diverse multimodal datasets, including image, text, video, audio, and cancer-related omics data. Our study establishes a novel theoretical framework for understanding cross-modal KD and offers practical guidelines based on the CCH criterion to select optimal teacher modalities for improving the performance of weaker modalities.

## 1 INTRODUCTION

Knowledge distillation (KD) transfers knowledge from a well-performing "teacher" model to a smaller, simpler "student" model in order to reduce computational costs at prediction time(Camilli et al., 2023; Maillard et al., 2024; Gou et al., 2021; Choi et al., 2023; Cheng et al., 2020; Huang et al., 2022; Tang et al., 2020). In standard KD, teacher and student networks have access to the same type of input data (Mishra and Marr, 2017); however, with the increasing availability of multimodal data, cross-modal KD has become increasingly popular (Liu et al., 2023).

Cross-modal KD enables a student network, typically operating on a less informative modality, to benefit from richer representations provided by a teacher network trained on a more informative modality (Gupta et al., 2016; Dai et al., 2021; Ahmad et al., 2024; Nair and Hänsch, 2024). Such methods are particularly valuable in scenarios where richer auxiliary modalities, such as video, audio, or text, are available during training, but only a single limited modality is accessible during testing (Du et al., 2021; Kim et al., 2024; Zhao et al., 2024; Radevski et al., 2022). Another prominent example is medical diagnostics, where costly procedures like tissue biopsies or genomic sequencing may be available for a subset of patients, while more standard analyses are available for much larger cohorts. Cross-modal KD in principle enables a teacher trained with these privileged datasets to effectively guide a student model that relies solely on routine inputs (Jiang et al., 2021; Zhang et al., 2023).

While attractive in principle, the theoretical foundations of cross-modal KD are still not well understood, and, alongside success stories, there are also reports of instances where cross-modal KD fails to improve or even degrades student performance (Croitoru et al., 2021; Lee et al., 2023). Previous research primarily attributes these negative effects to the modality gap, differences between modalities that obstruct knowledge transfer and result in misaligned supervisory signals (Yuzhe et al., 2024; Huo et al., 2024). Various approaches have

aimed to mitigate these issues through complex fusion strategies or bespoke loss functions (Thoker and Gall, 2019; Wang et al., 2023; Bano et al., 2024; Li et al., 2024), but the general applicability of these solutions remains unclear.

Theoretical studies on cross-modal KD have so far been limited. Vapnik and Vashist (2009) introduced "privileged information," a theoretical concept demonstrating that extra training-only data can improve model robustness. Building on this idea, Lopez-Paz et al. (2015) developed the "generalized distillation" framework, demonstrating that distilling knowledge from privileged information reduces the student's sample complexity and accelerates training convergence. More recently, Xue et al. (2023) empirically showed that the effectiveness of cross-modal KD significantly depends on the degree of label-relevant information shared between teacher and student modalities. Despite these insights, existing research has yet to determine a quantifiable criterion for successful cross-modal KD.

To address this gap, we introduce the Cross-modal Complementarity Hypothesis (CCH), a simple criterion based on mutual information which enables the user to *a priori* decide on whether cross-modal KD can be successful. We prove the validity of the CCH criterion in simplified scenarios, and test it empirically across a number of data sets. The primary contributions of this paper are as follows:

- Introduction of the Cross-modal Complementarity Hypothesis (CCH), proposing conditions under which cross-modal KD yields performance gains based on mutual information criteria.

- Proof of the validaty of the CCH criterion in the latent (jointly) Gaussian case.

- Extensive empirical validation through diverse experiments on multimodal datasets, including image, text, video, audio, and cancer-related omics data, confirming the practical utility of the proposed CCH criterion and providing actionable guidance for selecting effective teacher modalities.

## 2 RELATED WORK

### 2.1 UNIMODAL KD

KD is a powerful technique for transferring the detailed class information learned by a large teacher model to a smaller student model. Formally, consider a supervised $K$-class classification problem where both teacher and student classifiers receive the same input modality $X$ and produce logits over the $K$ classes. Let $z_{\theta_1}(X)$ and $z_{\theta_2}(X)$ denote the pre-softmax logits of the teacher and student, respectively. Given a temperature $T$, we define the softened outputs

$$f_{\theta_i}(X; T) = \text{softmax}\big(z_{\theta_i}(X)/T\big).$$

The student is trained to minimize a weighted combination of the cross-entropy loss with respect to the ground-truth labels $Y$ and the distillation loss:

$$\mathcal{L} = (1 - \lambda) \, \text{CE}\big(Y, \, f_{\theta_2}(X; 1)\big) + \lambda \, T^2 \, \text{KL}\big(f_{\theta_1}(X; T) \, \| \, f_{\theta_2}(X; T)\big), \tag{1}$$

where $\lambda \in [0, 1]$ balances learning directly from labels with learning from the teacher's predictions. The factor $T^2$ compensates for smaller gradients at higher temperatures, and the softened teacher outputs $f_{\theta_1}(X; T)$ convey richer inter-class relationships than one-hot labels alone (Hinton et al., 2015).

### 2.2 CROSS-MODAL KD

Cross-modal KD generalizes the unimodal framework to heterogeneous modalities, allowing a teacher with access to a stronger modality to guide a student with a weaker one. Consider two distinct modalities, denoted by $X_1$ and $X_2$, processed by the teacher and student models, respectively. The training objective extends Eq. equation 1 by appropriately substituting these distinct inputs (Liu et al., 2021):

$$\mathcal{L} = (1 - \lambda) \, \text{CE}\big(Y, \, f_{\theta_2}(X_2; 1)\big) + \lambda \, T^2 \, \text{KL}\big(f_{\theta_1}(X_1; T) \, \| \, f_{\theta_2}(X_2; T)\big). \tag{2}$$

**Modality gaps**   Cross-modal KD encounters substantial obstacles due to the inherent modality gap between the teacher and student data representations. These disparities arise because modalities like images, text, and audio capture and encode information through fundamentally distinct physical processes and mathematical formalisms (Hu et al., 2023; Sarkar and Etemad, 2024; Wang et al., 2025). Previous research indicates that modality gaps lead to both modality imbalance—the disparity in predictive power across modalities—and soft label misalignment—where the teacher's outputs do not align with the student's feature space. Consequently, these issues severely hinder effective knowledge transfer, thereby diminishing the efficacy of distillation (Huo et al., 2024). To mitigate these challenges, several studies have framed cross-modal KD as an information-maximization problem, proposing that effective transfer is achieved by maximizing the mutual information between the teacher's and student's representations or outputs (Ahn et al., 2019; Chen et al., 2021; Shrivastava et al., 2023; Xia et al., 2023; Shi et al., 2024; Li et al., 2024).

**Theoretical foundations**   Vapnik and Vashist (2009) introduced the concept of "privileged information" as data available only during training. This provides a theoretical reason why additional inputs—often from a different modality—can improve model robustness. This idea naturally applies to cross-modal transfer, where the teacher's modality acts as privileged information for the student. Building on this idea, later work Lopez-Paz et al. (2015) unified knowledge distillation with the privileged information framework, providing both theoretical and causal insights. Recent hypotheses further suggest that the success of cross-modal KD largely depends on the proportion of label-relevant information shared between teacher and student modalities (Xue et al., 2023). Another related hypothesis proposes that domain gaps mainly affect student performance through errors in non-target classes. Theoretical analyses based on VC theory show that reducing divergence in these off-target predictions improves student performance (Chen et al., 2024). Despite these advances, no previous work has explicitly defined conditions based on mutual information to determine when cross-modal KD is feasible.

## 3   THE CROSS-MODAL COMPLEMENTARITY HYPOTHESIS

We study cross-modal KD in settings where the teacher and student models access modalities of unequal predictive power. Let $X_1$ and $X_2$ denote two data modalities whose intrinsic capacities differ, and let $Y$ be the ground-truth label. Concretely, we assume $X_1$ to be the inputs to the teacher network, i.e. the data associated with the strong modality which is highly predictive of the output labels, while $X_2$ is the weak modality supplied to the student. The primary goal of cross-modal KD in this context is to transfer the label-relevant representations from the strong modality $X_1$ to the weak modality $X_2$, thereby augmenting the student's performance. This raises a fundamental question: under what conditions can a teacher operating on a strong modality effectively compensate for the insufficiencies of a weak modality?

Denote $H_1, H_2$ to be the represenation of $X_1, X_2$. Our intuition is that if the mutual information between $H_1$ and $H_2$, denoted by $I(H_1; H_2)$, exceeds the mutual information between $H_2$ and $Y$, denoted by $I(H_2; Y)$, the first term in contains more information than the second term, and thus the teacher modality $X_1$ can provide the complementary, label-relevant information that $X_2$ lacks. Also, a large $I(H_1; H_2)$ indicates substantial overlap between the modalities, suggesting that the student is capable of interpreting the teacher's guidance. This condition ensures that the teacher's knowledge is sufficiently aligned with the student's domain to improve prediction accuracy through distillation.

We thus propose the following *Cross-modal Complementarity Hypothesis*:

> **Cross-modal Complementarity Hypothesis (CCH):** For cross-modal knowledge distillation, if
> $$I(H_1; H_2) > I(H_2; Y),$$
> then the teacher modality can supply compensatory information, leading to improved student performance, where $H_1, H_2$ are teacher and student representations,

In the rest of this section, we support mathematically this intuition in a simple but tractable case.

Assume that the dataset $\{(x_{1i}, x_{2i}, y_i)\}_{i=1}^n$ is jointly Gaussian distributed:

$$\left\{ \begin{pmatrix} x_{1i} \\ x_{2i} \\ y_i \end{pmatrix} \right\}_{i=1}^n \overset{i.i.d.}{\sim} \mathcal{N}\left( 0, \begin{pmatrix} \Sigma_{11} & \Sigma_{12} & \Sigma_{13} \\ \Sigma_{12}^T & \Sigma_{22} & \Sigma_{23} \\ \Sigma_{13}^T & \Sigma_{23}^T & \Sigma_{33} \end{pmatrix} \right), \tag{3}$$

where $x_{1i}, x_{2i} \in \mathbb{R}^p$ and $y \in \mathbb{R}$. We consider the limit $n, p \to \infty$ with $\frac{n}{p} \to \kappa$ and the operation norm of each $\Sigma_{ij}$ ($1 \le i, j \le 3$) is bounded by a constant.

The associated learning task is a multi-modal (linear) regression problem with data $\mathcal{D} = \{x_{1i}, x_{2i}, y_i\}_{i=1}^n$. The outputs of the teacher and student networks for the $i$-th sample are $w_1^T x_{1i}$ and $w_2^T x_{2i}$, respectively, where $w_1$ and $w_2$ are the trainable parameters. The cross-modal objective for training the student is given by

$$\hat{w} := \arg\min_{w_2} \sum_{i=1}^n \left\| y_i - w_2^T x_{2i} \right\|^2 + \lambda \sum_{i=1}^n \left\| w_2^T x_{2i} - w_1^T x_{1i} \right\|^2, \tag{4}$$

where the first term measures the discrepancy between the ground-truth label and the student's predictions, and the second term, weighted by $\lambda$, enforces alignment between teacher and student outputs.

The excess risk is given by

$$R(\lambda, w_1) := \mathbb{E}_{x_1, x_2, y}[(y - (\hat{w})^T x_2)^2] - \sigma^2, \tag{5}$$

which is regarded as a function of the teacher weights $w_1$ (with bounded norm) and the regularization strength $\lambda$. We then define $R_0 := R(0, w_1)$ to be the baseline performance, where the teacher is absent and obviously $R_0$ does not depend on $w_1$. Then we have the following theorem.

**Theorem 1.** *Assume that $\kappa > 1$ and $w_1^T \Sigma_{11} w_1 \le \Sigma_{33}$, $w_1^T \Sigma_{13} \ge 0$. Suppose that $I(w_1^T x_1, (w^*)^T x_2) > I((w^*)^T x_2, y)$, where $w^* := \Sigma_{22}^{-1} \Sigma_{23}$ is the optimal student weight, then we have*

$$R(\lambda, w_1) < R_0 \tag{6}$$

*asymptotically for small $\lambda$.*

Note that $w_1^T \Sigma_{11} w_1 \le \Sigma_{33}$, $w_1^T \Sigma_{13} \ge 0$ are mild assumptions that the teacher weights should not be too large or too misleading. Notably the optimal teacher weight $\Sigma_{11}^{-1} \Sigma_{13}$ satisfies these two assumptions.

Theorem 1 suggests that knowledge distillation is beneficial when the mutual information between teacher and student representations are larger than the mutual information between student representations and the teacher. It is proved in Appendix A. *For the following we provide an explanation in a non-linear setting. The training objective for the student network is*

$$\sum_{n=1}^N \left\| y_n - f(w_2^T x_{2n}) \right\|^2 + \lambda \sum_{n=1}^N \left\| f(w_2^T x_{2n}) - f(w_1^T x_{1n}) \right\|^2, \tag{7}$$

*which can be equivalently expressed as*

$$\sum_{n=1}^N \left\| \tfrac{1}{1+\lambda}\left( y_n + \lambda f(w_1^T x_{1n}) \right) - f(w_2^T x_{2n}) \right\|^2.$$

*This formulation can be viewed as substituting the original label $y_n$ with the new label $\frac{1}{1+\lambda}\left( y_n + \lambda f(w_1^T x_{1n}) \right)$. This new label is more "accurate" if*

$$I\left( f(w_1^T X_1), \Sigma_{23}^T X_2 \right) \ge I\left( Y, \Sigma_{23}^T X_2 \right). \tag{8}$$

*By applying data processing inequalities, one obtains*

$$I\left( w_1^T X_1, \Sigma_{23}^T X_2 \right) \ge I\left( f(w_1^T X_1), \Sigma_{23}^T X_2 \right) \ge I\left( Y, \Sigma_{23}^T X_2 \right), \tag{9}$$

*which is the CCH criterion.*

## 4 EXPERIMENTS

To validate the proposed Cross-modal Complementarity Hypothesis (CCH), we conducted extensive experiments across various datasets, including synthetic data, image, text, video, audio, and cancer-related omics datasets. To systematically assess how mutual information influences the effectiveness of cross-modal KD, the teacher and student networks were intentionally configured to have identical architectures in all experiments. This design choice facilitates a clear and unbiased comparison, isolating mutual information as the primary variable affecting knowledge transfer effectiveness.

### 4.1 SYNTHETIC DATA

We generate synthetic data for a regression task by drawing $n$ i.i.d. samples from a zero-mean multivariate Gaussian model (cf. Eq. 3) over a teacher modality $X_1 \in \mathbb{R}^{n \times p}$, a student modality $X_2 \in \mathbb{R}^{n \times p}$, and a scalar target $Y \in \mathbb{R}^n$. To enable controlled analyses, we specialize the Gaussian model by parameterizing all cross-covariances as scalar multiples of the identity. Specifically,

$$\Sigma_{12} = \sigma_{12}I_p, \quad \Sigma_{13} = \sigma_{13}I_p, \quad \Sigma_{23} = \sigma_{23}I_p, \quad \mathrm{Var}(Y) = 1,$$

where each $\sigma_{ij} \in (-1, 1)$ governs the corresponding pairwise correlation. Under this parameterization,

$$\left\{ \begin{pmatrix} x_{1i} \\ x_{2i} \\ y_i \end{pmatrix} \right\}_{i=1}^{n} \sim \mathcal{N}\left( 0, \begin{pmatrix} I_p & \sigma_{12}I_p & \sigma_{13}\mathbf{1}_p \\ \sigma_{12}I_p & I_p & \sigma_{23}\mathbf{1}_p \\ \sigma_{13}\mathbf{1}_p^\top & \sigma_{23}\mathbf{1}_p^\top & 1 \end{pmatrix} \right), \tag{10}$$

so that $I(X_1; X_2)$, $I(X_1; Y)$, and $I(X_2; Y)$ are monotone in $\sigma_{12}$, $\sigma_{13}$, and $\sigma_{23}$, respectively.

Unless otherwise stated, we set $n = 10000$ and $p = 100$. To study how student performance varies with cross-modal dependence, we fix the teacher–label correlation at $\sigma_{13} = 0.9$ and the student–label correlation at $\sigma_{23} = 0.4$, and vary $\sigma_{12} \in [0, 0.7]$ to maintain positive semidefiniteness of the covariance.

Figure 1 summarizes the results. Panel 1a reports the student test mean squared error (MSE) as $\sigma_{12}$ varies; each point averages ten random seeds. Panel 1b shows mutual information (MI) between learned representations: $I(H_1; H_2)$ for teacher $X_1$ and student $X_2$, and $I(H_2; Y)$ for the student and the label. We extract representations $H_1$ and $H_2$ from each network's feature extractor and estimate MI using the `latentmi` estimator (Gowri et al., 2024).

Empirically, knowledge distillation (KD) reduces MSE precisely when $I(H_1; H_2) > I(H_2; Y)$ and provides no benefit otherwise. This pattern supports the Cross-modal Complementarity Hypothesis (CCH): the teacher contributes complementary, label-relevant information when its representation shares more information with the student than the student shares with the label. Additional experiments across distillation weights $\lambda$ (Appendix B) corroborate this trend.

### 4.2 IMAGE DATA

We conduct classification experiments on the MNIST (LeCun et al., 1998) and MNIST-M datasets (Ganin and Lempitsky, 2015). MNIST is a standard benchmark of 70,000 handwritten digits (0–9), each a $28 \times 28$-pixel grayscale image with a corresponding label. MNIST-M is derived by blending the binarized MNIST digits onto random natural-image patches from the BSDS500 dataset (Martin et al., 2001); thus, it represents a distinct modality while sharing identical labels with MNIST (see Figure 6 in Appendix C).

We treat MNIST as the *teacher* modality and MNIST-M as the *student* modality. First, we compute the mutual information between the teacher and student representations, $I_{TS} = I\big(H_{\mathrm{MNIST}}; H_{\mathrm{MNIST\text{-}M}}\big)$, and between the student represntations and labels, $I_{SL} = I\big(H_{\mathrm{MNIST\text{-}M}}; Y\big)$, using the `latentmi` estimator (Gowri et al., 2024). We then follow the protocol in Algorithm 1 (Appendix C). During distillation, we systematically vary $I_{TS}$ by applying Gaussian blur with standard deviation $\gamma$ to the teacher inputs, and assess whether the student's accuracy gains correspond to the CCH condition $I_{TS} > I_{SL}$.

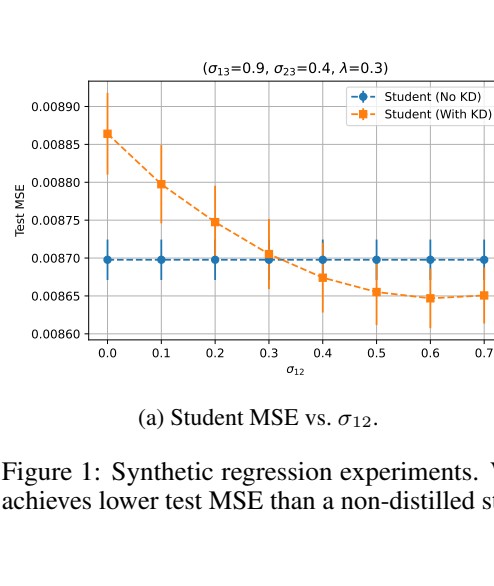

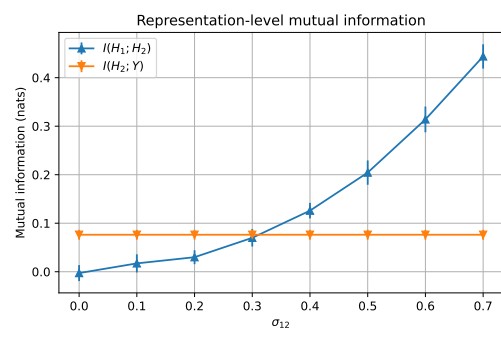

(a) Student MSE vs. $\sigma_{12}$.

(b) Representation MI vs. $\sigma_{12}$.

Figure 1: Synthetic regression experiments. When $I(H_1; H_2)$ exceeds $I(H_2; Y)$, the KD-trained student achieves lower test MSE than a non-distilled student; otherwise, KD provides no improvement.

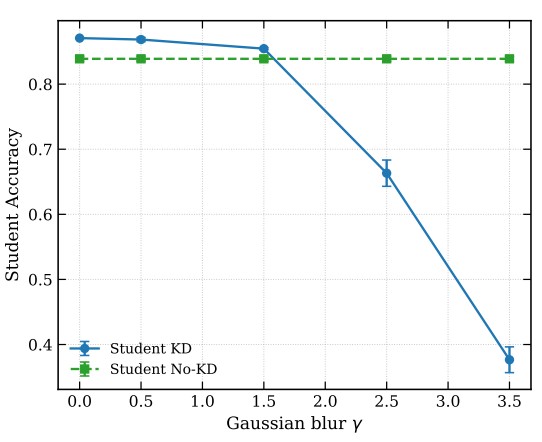

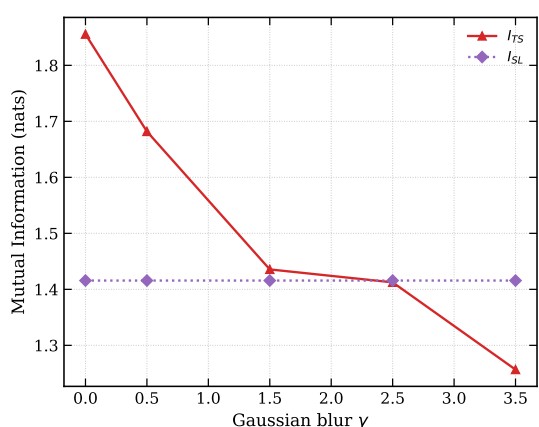

(a) Student accuracy vs. Gaussian blur $\gamma$.

(b) Mutual information vs. Gaussian blur $\gamma$.

Figure 2: Relationship between student accuracy and mutual information under varying Gaussian blur. (**a**) Test accuracy of the MNIST–M student trained with (solid line) and without (dashed line) distillation as a function of Gaussian blur standard deviation $\gamma$ applied to MNIST teacher inputs. (**b**) Mutual information $I_{TS} = I(H_{\text{MNIST}}; H_{\text{MNIST-M}})$ (red) and $I_{SL} = I(H_{\text{MNIST-M}}; Y)$ (purple) versus $\gamma$. Accuracy improvements align with the region where $I_{TS} > I_{SL}$. For reference, $I_{TL} = I(H_{\text{MNIST}}; Y) = 2.0485$, and the teacher network attains a test accuracy of 0.981.

Figure 2 illustrates the impact of varying Gaussian blur intensity $\gamma$ on both the student's test accuracy and the corresponding mutual information when the distillation temperature is at $T = 3$ (see additional results in Appendix C). Results are averaged over five independent runs. Panel (a) compares the test accuracy of students trained with and without distillation; panel (b) plots $I_{TS}$ and $I_{SL}$ as functions of $\gamma$. We observe that whenever $I_{TS} > I_{SL}$, knowledge distillation improves accuracy relative to the baseline, in agreement with the CCH. For $\gamma \geq 2.5$, $I_{TS}$ falls below $I_{SL}$, leading to a collapse in the distilled student's performance.

Table 1: Mutual-information gap and student accuracy differ under varying blur and temperature.

| $\gamma$ | MI GAP (nats) | Student Acc. Diff. ($\pm$SE) | | | |
|---|---|---|---|---|---|
| | | $T = 1$ | $T = 2$ | $T = 3$ | $T = 4$ |
| 0.0 | 0.4399 | $0.0010 \pm 0.0040$ | $0.0146 \pm 0.0035$ | $0.0318 \pm 0.0040$ | $0.0350 \pm 0.0046$ |
| 0.5 | 0.2662 | $0.0069 \pm 0.0054$ | $0.0152 \pm 0.0055$ | $0.0296 \pm 0.0031$ | $0.0353 \pm 0.0028$ |
| 1.5 | 0.0199 | $0.0002 \pm 0.0089$ | $0.0149 \pm 0.0034$ | $0.0156 \pm 0.0042$ | $0.0091 \pm 0.0051$ |
| 2.5 | $-0.0032$ | $-0.1190 \pm 0.0165$ | $-0.1627 \pm 0.0101$ | $-0.1757 \pm 0.0219$ | $-0.1516 \pm 0.0154$ |
| 3.5 | $-0.1590$ | $-0.2797 \pm 0.0126$ | $-0.4597 \pm 0.0041$ | $-0.4623 \pm 0.0209$ | $-0.4364 \pm 0.0137$ |

Table 2: Mutual information estimates between CMU-MOSEI modality representations and the label using three estimators (mean $\pm$ std over 50 runs).

| Estimator | $I(H_{\text{text}}; H_{\text{vision}})$ | $I(H_{\text{text}}; H_{\text{audio}})$ | $I(H_{\text{text}}; Y)$ | $I(H_{\text{vision}}; Y)$ | $I(H_{\text{audio}}; Y)$ |
|---|---|---|---|---|---|
| latentmi | $1.3543 \pm 0.0052$ | $1.4160 \pm 0.0038$ | $0.4681 \pm 0.0090$ | $0.0816 \pm 0.0084$ | $0.1054 \pm 0.0088$ |
| mine | $0.7955 \pm 0.0019$ | $1.1817 \pm 0.0023$ | $0.3202 \pm 0.0055$ | $0.0409 \pm 0.0026$ | $0.0631 \pm 0.0026$ |
| ksg | $0.3788 \pm 0.0056$ | $0.6606 \pm 0.0056$ | $0.1628 \pm 0.0083$ | $0.0647 \pm 0.0014$ | $0.0934 \pm 0.0018$ |

Table 3: Student performance versus mutual information on CMU-MOSEI with text as teacher. The teacher achieves test accuracy $0.7190 \pm 0.0098$ and weighted F1 $0.7189 \pm 0.0098$; $I(H_{\text{text}}; Y) = 0.4681 \pm 0.0090$. Mutual information is estimated with latentmi.

| | $I(H_{\text{teacher}}; H_{\text{student}})$ | $I(H_{\text{student}}; Y)$ | Student Without KD | | Student With KD | |
|---|---|---|---|---|---|---|
| | | | Acc | Weighted F1 | Acc | Weighted F1 |
| Text (teacher) Vision (student) | $1.3543 \pm 0.0052$ | $0.0816 \pm 0.0084$ | $0.6233 \pm 0.0027$ | $0.6204 \pm 0.0030$ | $0.6343 \pm 0.0013$ | $0.6315 \pm 0.0022$ |
| Text (teacher) Audio (student) | $1.4160 \pm 0.0038$ | $0.1054 \pm 0.0088$ | $0.5937 \pm 0.0048$ | $0.5931 \pm 0.0043$ | $0.6167 \pm 0.0030$ | $0.6161 \pm 0.0031$ |

We further explore the effect of the distillation temperature $T \in \{1, 2, 3, 4\}$ in Table 1. Here, *MI GAP* denotes $I_{TS} - I_{SL}$, and *Student Acc. Diff.* is the difference in test accuracy between the distilled and baseline students. SE denotes the standard error estimated from five independent runs. Across all blur levels and temperatures, the sign of the *Student Acc. Diff.* matches that of the *MI GAP*, reinforcing the CCH. We remark the very non-linear behaviour of the student's accuracy w.r.t. the MI GAP; while the gain remains modest for positive MI GAP, as soon as the MI GAP changes sign we document a very large drop in student accuracy.

## 4.3 CMU-MOSEI DATASET

We evaluate the CCH on the CMU Multimodal Opinion Sentiment and Emotion Intensity (CMU-MOSEI) dataset (Zadeh et al., 2018). CMU-MOSEI is a large-scale benchmark for multimodal sentiment analysis comprising 23,453 annotated video segments with time-aligned text, vision, and audio streams drawn from 1,000 speakers across 250 topics.

The task is binary sentiment classification. Following standard practice, we binarize the original integer sentiment scores into positive and negative labels. Each utterance is converted into synchronized, fixed-length sequences for all three modalities using a uniform preprocessing pipeline; full details are provided in Appendix D.

To operationalize the CCH, we estimate mutual information (MI) between (i) each pair of modality representations and (ii) each modality representation and the label. We employ three complementary estima-

Table 4: Student weighted F1 versus mutual information on the CMU-MOSEI dataset under varying levels of Gaussian noise (text teacher, vision student).

| Noise level | $I(H_{\text{teacher}}; H_{\text{student}})$ | $I(H_{\text{student}}; Y)$ | Student KD F1 | Student No-KD F1 |
|---|---|---|---|---|
| 0% | $1.3543 \pm 0.0052$ | $0.0816 \pm 0.0084$ | $0.6204 \pm 0.0030$ | $0.6315 \pm 0.0022$ |
| 20% | $0.0034 \pm 0.0040$ | $0.0816 \pm 0.0084$ | $0.6204 \pm 0.0030$ | $0.6192 \pm 0.0062$ |
| 40% | $-0.0007 \pm 0.0045$ | $0.0816 \pm 0.0084$ | $0.6204 \pm 0.0030$ | $0.6189 \pm 0.0039$ |
| 60% | $-0.0056 \pm 0.0058$ | $0.0816 \pm 0.0084$ | $0.6204 \pm 0.0030$ | $0.6184 \pm 0.0022$ |
| 80% | $-0.0060 \pm 0.0053$ | $0.0816 \pm 0.0084$ | $0.6204 \pm 0.0030$ | $0.6156 \pm 0.0033$ |

Table 5: Student weighted F1 vs. mutual information on BRCA under varying Gaussian noise levels (*teacher:* mRNA; *student:* CNV). The teacher achieves test weighted F1 of 0.7459 and $I(H_{\text{teacher}}; Y) = 1.1081$. "MI Gap" denotes $I_{\text{TS}} - I_{\text{SL}}$; "Student F1 Difference" denotes (Student KD F1) − (Student No-KD F1).

| Noise Level | $I(H_{\text{teacher}}; H_{\text{student}})$ | $I(H_{\text{student}}; Y)$ | Student KD F1 | Student No-KD F1 | MI GAP | Student F1 Differ |
|---|---|---|---|---|---|---|
| 0% | 0.5005 | 0.2757 | 0.5038 | 0.4561 | 0.2248 | 0.0477 |
| 20% | 0.4554 | 0.2757 | 0.4917 | 0.4561 | 0.1797 | 0.0356 |
| 40% | 0.3687 | 0.2757 | 0.4953 | 0.4561 | 0.0930 | 0.0392 |
| 60% | 0.2147 | 0.2757 | 0.4276 | 0.4561 | -0.061 | -0.0285 |
| 80% | 0.1325 | 0.2757 | 0.4343 | 0.4561 | -0.1432 | -0.0218 |

tors—`latentmi` (Gowri et al., 2024), `mine` (Belghazi et al., 2018), and `ksg` (Ross, 2014)—and average results over 50 independent runs (Appendix F). As shown in Table 2, absolute MI values vary by estimator, but the relative ordering is consistent.

The MI patterns in Table 2 identify text as the most predictive modality, since $I(H_{\text{text}}; Y)$ is largest. Accordingly, we designate text as the teacher and treat vision and audio as student modalities. As reported in Table 3, KD yields significant gains over the no-KD baseline for both students. Moreover, Table 2 shows that $I(H_{\text{text}}; H_{\text{vision}}) > I(H_{\text{vision}}; Y)$ and $I(H_{\text{text}}; H_{\text{audio}}) > I(H_{\text{audio}}; Y)$, satisfying the CCH condition. Taken together, these observations support the CCH. The improvement is larger for audio, consistent with its greater MI gap $I_{\text{TS}} - I_{\text{SL}}$ (teacher–student vs. student–label MI of representations), suggesting a positive association between the gap magnitude and KD efficacy.

To further probe the CCH, we conduct a controlled degradation experiment on the text (teacher) →vision (student) setting. We inject Gaussian noise into the teacher input to systematically reduce $I(H_{\text{teacher}}; H_{\text{student}})$ while holding $I(H_{\text{student}}; Y)$ fixed. As predicted, the benefit of KD disappears once $I(H_{\text{teacher}}; H_{\text{student}}) < I(H_{\text{student}}; Y)$ (Table 4).

## 4.4 CANCER DATA

We analyze three The Cancer Genome Atlas (TCGA) cohorts (Colaprico et al., 2016): breast invasive carcinoma (BRCA), pan-kidney (KIPAN), and liver hepatocellular carcinoma (LIHC). For each cohort, we consider three omics modalities—mRNA expression (mRNA), copy number variation (CNV), and reverse-phase protein arrays (RPPA)—and retain only cases with complete data across all three. The learning task is subtype classification; Table 19 in Appendix E reports class distributions. To reduce noise and dimensionality, we preprocess each modality independently and select the top 100 features from the original sets of 60,660 (mRNA), 60,623 (CNV), and 487 (RPPA) using the minimum-redundancy maximum-relevance (mRMR) criterion (Ding and Peng, 2005).

We first set mRNA as the teacher and CNV as the student and estimate

$$I_{\text{TS}} = I(H_{\text{mRNA}}; H_{\text{CNV}}), \qquad I_{\text{SL}} = I(H_{\text{CNV}}; Y),$$

Table 6: Student weighted F1 vs. mutual information on KIPAN under varying Gaussian noise levels (*teacher:* mRNA; *student:* CNV). The teacher achieves test weighted F1 of $0.9516$ and $I(H_{\text{teacher}}; Y) = 1.0458$.

| Noise Level | $I(H_{\text{teacher}}; H_{\text{student}})$ | $I(H_{\text{student}}; Y)$ | Student KD F1 | Student No-KD F1 | MI GAP | Student F1 Differ |
|---|---|---|---|---|---|---|
| 0% | 0.7898 | 0.6994 | 0.8826 | 0.8667 | 0.0904 | 0.0159 |
| 20% | 0.7198 | 0.6994 | 0.8721 | 0.8667 | 0.0204 | 0.0054 |
| 40% | 0.6771 | 0.6994 | 0.8517 | 0.8667 | -0.0223 | -0.0150 |
| 60% | 0.6209 | 0.6994 | 0.8477 | 0.8667 | -0.0785 | -0.0190 |
| 80% | 0.6389 | 0.6994 | 0.8544 | 0.8667 | -0.0605 | -0.0123 |

Table 7: Student weighted F1 vs. mutual information on LIHC under varying Gaussian noise levels (*teacher:* mRNA; *student:* CNV). The teacher achieves test weighted F1 of $0.9430$ and $I(H_{\text{teacher}}; Y) = 0.9055$.

| Noise Level | $I(H_{\text{teacher}}; H_{\text{student}})$ | $I(H_{\text{student}}; Y)$ | Student KD F1 | Student No-KD F1 | MI GAP | Student F1 Differ |
|---|---|---|---|---|---|---|
| 0% | 0.0914 | 0.0781 | 0.5795 | 0.5548 | 0.0133 | 0.0247 |
| 20% | 0.0825 | 0.0781 | 0.5692 | 0.5548 | 0.0044 | 0.0144 |
| 40% | 0.0699 | 0.0781 | 0.5368 | 0.5548 | -0.0082 | -0.0180 |
| 60% | 0.0736 | 0.0781 | 0.5259 | 0.5548 | -0.0045 | -0.0289 |
| 80% | 0.0409 | 0.0781 | 0.5080 | 0.5548 | -0.0372 | -0.0468 |

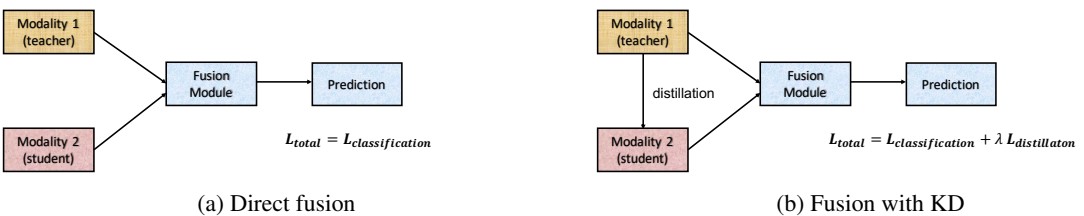

(a) Direct fusion          (b) Fusion with KD

Figure 3: Multimodal fusion architectures: direct fusion (left) and Fusion+KD (right).

Table 8: Overall multimodal performance of direct fusion and Fusion+KD on KIPAN, reported with mutual information of modality representations (teacher–label, teacher–student, student–label).

| | Mutual Information | | | Fusion | | | | Fusion+KD | | | |
|---|---|---|---|---|---|---|---|---|---|---|---|
| | Teacher–Label | Teacher–Student | Student–Label | Acc | AUC | Macro F1 | Weighted F1 | Acc | AUC | Macro F1 | Weighted F1 |
| mRNA (teacher) CNV (student) | 1.0458 | 0.7898 | 0.6994 | 0.9610 | 0.9851 | 0.9219 | 0.9591 | 0.9740 | 0.9872 | 0.9293 | 0.9725 |
| RPPA (teacher) CNV (student) | 1.1609 | 0.6893 | 0.6994 | 0.9740 | 0.9995 | 0.9333 | 0.9721 | 0.9610 | 0.9971 | 0.9225 | 0.9595 |

using the `latentmi` estimator. To modulate $I_{\text{TS}}$, we add zero-mean Gaussian noise to the teacher inputs. Tables 5–7 report student weighted F1 and mutual information as functions of the noise level (means over five runs). Across cohorts, whenever the MI Gap is positive ($I_{\text{TS}} > I_{\text{SL}}$), distillation improves the student's weighted F1; when the gap becomes negative, the benefit vanishes or reverses, in line with the CCH.

To extend from single-student distillation to multimodal learning, we compare two fusion strategies—direct fusion and fusion with knowledge distillation (Fusion+KD; Fig. 3). On KIPAN (Table 8; additional results in Appendix E), mRNA as teacher yields $I_{\text{TS}} > I_{\text{SL}}$ and Fusion+KD outperforms direct fusion. In contrast, with RPPA as teacher we have $I_{\text{TS}} < I_{\text{SL}}$, and direct fusion is superior. These results suggest a practical design rule: incorporate KD in fusion only when $I_{\text{TS}} > I_{\text{SL}}$.

## 5    CONCLUSION

This paper introduced the Cross-modal Complementarity Hypothesis (CCH), a framework for explaining when cross-modal knowledge distillation (KD) improves performance in multimodal learning. The CCH offers a tractable, *a priori* criterion for success: distillation is beneficial when the mutual information between teacher and student representations exceeds that between the student representation and the labels. We validated the hypothesis with a theoretical analysis in a joint Gaussian model and with experiments spanning synthetic Gaussian data and diverse real-world modalities—image, text, video, and audio—as well as three cancer omics datasets.

Our results highlight mutual information as a reliable predictor of cross-modal KD efficacy, yielding both theoretical insight and practical guidance for selecting teacher modalities to strengthen weaker ones.

## REPRODUCIBILITY STATEMENT

The source code underpinning the experiments and analyses presented in this manuscript has been made accessible via an anonymized GitHub repository:

$$\texttt{https://anonymous.4open.science/r/test-111/.}$$

All experiment details are presented in Appendices B-F.

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

## A  THEORETICAL ANALYSIS

Here we prove a more complete version of Theorem 1.

**Theorem 2.** *For $\kappa > 1$ and almost every $\lambda$, there exists $w_1$ such that $R(\lambda, \tilde{w}) < R(\lambda, 0)$ asymptotically. Moreover, for $\lambda$ small enough, we have $R(\lambda, \tilde{w}) < R_0$ asymptotically if $w_1^T \Sigma_{11} w_1 \le \Sigma_{33}, w_1^T \Sigma_{13} \ge 0$ and $I(w_1^T x_1, (w^*)^T x_2) > I((w^*)^T x_2, y)$.*

*Proof.* The optimization problem eq. (4) is equivalent to

$$\hat{w} := \arg\min_{w_2} \sum_{i=1}^{n} \left\| \tilde{y}_i - w_2^T x_{2i} \right\|^2, \tag{11}$$

where the effective label is given by

$$\bar{y}_i := \frac{1}{1+\lambda}(y_i + \lambda w_1^T x_{1i}). \tag{12}$$

It satisfies $\bar{y}_i = \bar{w}^T x_{2i} + \mathcal{N}(0, \bar{\sigma}^2)$, where

$$\bar{w} := \frac{1}{1+\lambda}\Sigma_{22}^{-1}(\Sigma_{23} + \lambda \Sigma_{12}^T w_1) \tag{13}$$

and

$$\tilde{\sigma}^2 := \mathbb{E}[\bar{y}_n^2] - \bar{w}^T \Sigma_{22} \bar{w}. \tag{14}$$

According to Theorem 3 of Chang et al. (2021), the estimator $\hat{w}$ can be expressed asymptotically as

$$\hat{w} = \bar{w} + \bar{\sigma}\frac{\Sigma_{22}^{-1/2} g}{\sqrt{p(\kappa - 1)}}, \tag{15}$$

where $g \sim \mathcal{N}(0, I_p)$. Thus the asymptotics of $R(\lambda, w_1)$ is

$$\bar{R}(\lambda, w_1) = (\bar{w} - w^*)\Sigma_{22}(\bar{w} - w^*) + \tilde{\sigma}^2 \frac{1}{\kappa - 1}$$

$$= \frac{\lambda^2}{(1+\lambda)^2}(\Sigma_{22}^{-1}\Sigma_{12}^T w_1 - w^*)^T \Sigma_{22}(\Sigma_{22}^{-1}\Sigma_{12}^T w_1 - w^*)$$

$$+ \frac{1}{\kappa - 1}\frac{1}{(1+\lambda)^2}[\Sigma_{33} - (w^*)^T \Sigma_{22} w^* + 2\lambda w_1^T(\Sigma_{13} - \Sigma_{12}w^*) + \lambda^2 w_1^T(\Sigma_{11} - \Sigma_{12}\Sigma_{22}^{-1}\Sigma_{12}^T)w_1], \tag{16}$$

where we denote $w^* = \Sigma_{22}^{-1}\Sigma_{23}$ to be the optimal weight. Here "asymptotics" means that $\lim_{n,p\to\infty} \mathbb{P}\left(\sup_{||w_1||<M} |R(\lambda, w_1) - \bar{R}(\lambda, w_1)| > \epsilon\right) = 0$ for any $\epsilon > 0$. Taking the derivative of $\bar{R}$ w.r.t. $w_1$, we have that the optimal $w_1$ is given by

$$\lambda \left[\Sigma_{12}^T \Sigma_{22}^{-1}\Sigma_{12} + \frac{1}{\kappa - 1}(\Sigma_{11} - \Sigma_{12}\Sigma_{22}^{-1}\Sigma_{12}^T)\right] w_1 = \lambda \Sigma_{12}^T w^* - \frac{1}{\kappa - 1}(\Sigma_{13} - \Sigma_{12}w^*). \tag{17}$$

This gives an optimal $w_1$ for almost every $\lambda$. The optimal $w_1$ is non-zero and different from the optimal teacher weight $w^*$ for almost every $\lambda$. For the special case $\Sigma_{13} - \Sigma_{12}\Sigma_{22}^{-1}\Sigma_{23} = 0$ (i.e. $x_1$ and $y$ are independent conditioned on $x_2$), the optimal surrogate weight is given by

$$w_1 = (\kappa - 1)(\Sigma_{11} + (\kappa - 2)\Sigma_{12}\Sigma_{22}^{-1}\Sigma_{12}^T)^{-1}\Sigma_{12}^T w^*, \tag{18}$$

which does not depend on $\lambda$.

Moreover, for small $\lambda$, we have

$$\bar{R}(\lambda, w_1) = \frac{1}{\kappa - 1}\frac{1}{(1+\lambda)^2}(\Sigma_{33} - (w^*)^T\Sigma_{22}w^*) + \frac{2\lambda}{\kappa - 1}w_1^T(\Sigma_{13} - \Sigma_{12}w^*) + O(\lambda^2), \quad (19)$$

and thus $\bar{R}(\lambda, w_1) < \bar{R}(0, w_1)$ for small $\lambda$ if

$$\hat{w}^T(\Sigma_{13} - \Sigma_{12}\Sigma_{22}^{-1}\Sigma_{23}) - (\Sigma_{33} - (w^*)^T\Sigma_{22}w^*) < 0. \quad (20)$$

Now we define the correlation between $w_1 x_1$ and $w^* x_2$ to be

$$\rho(w_1 x_1, w^* x_2) := \frac{w_1^T\Sigma_{12}w^*}{\sqrt{w_1^T\Sigma_{11}w_1}\sqrt{(w^*)^T\Sigma_{22}w^*}}. \quad (21)$$

Similarly we define

$$\rho(w_1 x_1, y) := \frac{w_1^T\Sigma_{13}}{\sqrt{w_1^T\Sigma_{11}w_1}\sqrt{(w^*)^T\Sigma_{22}w^*}} \quad (22)$$

and

$$\rho(w^* x_2, y) := \frac{(w^*)^T\Sigma_{23}}{\sqrt{(w^*)^T\Sigma_{22}w^*}\sqrt{\Sigma_{33}}} = \frac{\sqrt{(w^*)^T\Sigma_{22}w^*}}{\sqrt{\Sigma_{33}}}. \quad (23)$$

Then the condition equation 20 becomes

$$\rho(w_1 x_1, w^* x_2) > \frac{\rho(w_1 x_1, y)}{\rho(w^* x_2, y)} - \frac{1 - \rho(w^* x_2, y)^2}{\rho(w^* x_2, y)}\frac{\sqrt{\Sigma_{33}}}{\sqrt{w_1^T\Sigma_{11}w_1}}. \quad (24)$$

Therefore, if $I(w_1^T x_1, (w^*)^T x_2) > I((w^*)^T x_2, y)$ we have

$$\begin{aligned}
\rho(w_1 x_1, w^* x_2) > \rho(w^* x_2, y) &= \frac{1}{\rho(w^* x_2, y)} - \frac{1 - \rho(w^* x_2, y)^2}{\rho(w^* x_2, y)} \\
&\geq \frac{\rho(w_1 x_1, y)}{\rho(w^* x_2, y)} - \frac{1 - \rho(w^* x_2, y)^2}{\rho(w^* x_2, y)}\frac{\sqrt{\Sigma_{33}}}{\sqrt{w_1^T\Sigma_{11}w_1}}.
\end{aligned} \quad (25)$$

Thus the condition equation 20 is satisfied and we have $\bar{R}(\lambda, w_1) < \bar{R}(0, w_1)$. For the first inequality we use $I(A, B) = -\frac{1}{2}\log(1 - \rho(A, B)^2)$ for Gaussian variables $A, B$ and the fact that $\rho(w^* x_2, y), \rho(w_1 x_1, y) \geq 0$ if $w_1^T\Sigma_{13} \geq 0$. The last inequality is from $\rho(w_1 x_1, y) \leq 1$ and $\frac{\sqrt{\Sigma_{33}}}{\sqrt{w_1^T\Sigma_{11}w_1}} \leq 1$. This finishes the proof. $\square$

For completeness we also prove that knowledge distillation might help in the overparameterization regime.

**Theorem 3.** *For $\kappa < 1$ and almost every $\lambda$, there also exists $w_1$ such that $R(\lambda, w_1) < R(\lambda, 0)$ asymptotically.*

*Proof.* For $\kappa < 1$ we are in the overparameterization case and thus we consider the minimal norm estimator

$$\hat{w} = \arg\min_w \left\{ ||w|| : \sum_{i=1}^n ||\frac{1}{1+\lambda}(y_i + \lambda w_1^T x_{1i}) - w^T x_{2i}||^2 = 0 \right\}. \quad (26)$$

We can rewrite it as

$$\hat{w} = \frac{\bar{\sigma}}{\sigma}\arg\min_w \left\{ ||w|| : \sum_{i=1}^n ||\frac{\sigma}{\bar{\sigma}}\bar{y}_i - w^T x_{2i}||^2 = 0 \right\}, \quad (27)$$

where we recall that the effective label satisfies $\frac{\sigma}{\bar{\sigma}}\bar{y}_i = \frac{\sigma}{\bar{\sigma}}\bar{w}^T x_{2i} + \mathcal{N}(0, \sigma^2)$.

Then we can use (Ildiz et al., 2024, Theorem 4) for the function $f(x) = ||\Sigma_{22}^{1/2}(\frac{\bar{\sigma}}{\sigma}x - w^*)||^2$ to obtain the following asymptotic excess risk

$$
\begin{aligned}
\bar{R}(\lambda, w_1) =& (w_s - w^*)^T\theta_1^T\Sigma_{22}\theta_1(w_s - w^*) + \gamma(w^s)\mathbb{E}_{g_t}[\theta_2^T\Sigma_{22}\theta_2] \\
& + w^*(I - \theta_1)^T\Sigma_{22}(I - \theta_1)w^* - 2(w^*)^T(I - \theta_1)^T\Sigma_{22}\theta_1(w_s - w^*),
\end{aligned}
\tag{28}
$$

where we denote $w_s := \frac{\sigma}{\bar{\sigma}}\bar{w}$ and $\tau$ to be the solution of $\kappa = \frac{1}{p}\text{tr}((\Sigma_{22} + \tau I)^{-1}\Sigma_{22})$,

$$
\theta_1 := \frac{\bar{\sigma}}{\sigma}(\Sigma_{22} + \tau I)^{-1}\Sigma_{22}, \ \theta_2 := \frac{\bar{\sigma}}{\sigma}(\Sigma_{22} + \tau I)^{-1}\Sigma_{22}^{1/2}\frac{g_t}{\sqrt{p}},
\tag{29}
$$

and $g_t \sim \mathcal{N}(0, I_p)$. Moreover, $\gamma(w_s)$ is given by

$$
\gamma^2(w_s) = \kappa^{-1}\frac{\sigma^2 + \tau^2||\Sigma_{22}^{1/2}(\Sigma_{22} + \tau I)^{-1}w_s||^2}{1 - \frac{1}{n}\text{tr}((\Sigma_{22} + \tau I)^{-2}\Sigma_{22}^2)}.
\tag{30}
$$

The results can be simplified to

$$
\begin{aligned}
\bar{R}(\lambda, w_1) =& \frac{\bar{\sigma}^2}{\sigma^2}(w_s - w^*)\Sigma_{22}^3(\Sigma_{22} + \tau I)^{-2}(w_s - w^*) + \frac{\bar{\sigma}^2}{\sigma^2}\Omega\frac{\sigma^2 + \tau^2||\Sigma_{22}^{1/2}(\Sigma_{22} + \tau I)^{-1}w_s||^2}{1 - \Omega} \\
& - 2\frac{\bar{\sigma}}{\sigma}(w^*)^T\Sigma_{22}^2(\Sigma_{22} + \tau I)^{-2}(\Sigma_{22} + \tau I - \frac{\bar{\sigma}}{\sigma}\Sigma_{22})(w_s - w^*) \\
& + w^*(\Sigma_{22} + \tau I - \frac{\bar{\sigma}}{\sigma}\Sigma_{22})^2(\Sigma_{22} + \tau I)^{-2}\Sigma_{22}w^*,
\end{aligned}
\tag{31}
$$

where we denote $\Omega := \frac{1}{n}\text{tr}((\Sigma_{22} + \tau I)^{-2}\Sigma_{22}^2)$. Therefore, the optimal $w_1$ is given by the saddle points of equation 31, where

$$
w_s := \frac{\sigma}{(1 + \lambda)\bar{\sigma}}(w^* + \lambda\Sigma_{22}^{-1}\Sigma_{12}^Tw_1)
\tag{32}
$$

and

$$
\bar{\sigma} := \frac{1}{1 + \lambda}\sqrt{\sigma^2 + 2\lambda w_1^T(\Sigma_{13} - \Sigma_{12}w^*) + \lambda^2 w_1^T(\Sigma_{11} - \Sigma_{12}\Sigma_{22}^{-1}\Sigma_{12}^T)w_1}.
\tag{33}
$$

$\square$

## B EXPERIMENTAL DETAILS AND RESULTS FOR SYNTHETIC DATA

We evaluate the Cross-modal Complementarity Hypothesis (CCH) on a controlled synthetic regression benchmark. We generate $n$ i.i.d. samples $\{(X_{1,i}, X_{2,i}, Y_i)\}_{i=1}^n$ as follows:

$$
\begin{aligned}
Y_i &\sim \mathcal{N}(0, 1), \\
X_{2,i} \mid Y_i &\sim \mathcal{N}(\sigma_{23}Y_i\,\mathbf{1}_p, \ (1 - \sigma_{23}^2)I_p), \\
X_{1,i} \mid X_{2,i}, Y_i &\sim \mathcal{N}(a\,X_{2,i} + b\,Y_i, \ v\,I_p),
\end{aligned}
$$

where

$$
\phi = 1 - \sigma_{23}^2, \quad a = \frac{\sigma_{12} - \sigma_{13}\sigma_{23}}{\phi}, \quad b = \frac{\sigma_{13} - \sigma_{12}\sigma_{23}}{\phi}, \quad v = 1 - \frac{\sigma_{12}^2 + \sigma_{13}^2 - 2\,\sigma_{12}\sigma_{13}\sigma_{23}}{\phi}.
$$

Both teacher and student use the fully connected architecture in Table 9. We train on 10000 samples and hold out 1000 for testing. Models are optimized with Adam (learning rate 0.01) for 300 epochs; full settings appear in Table 10.

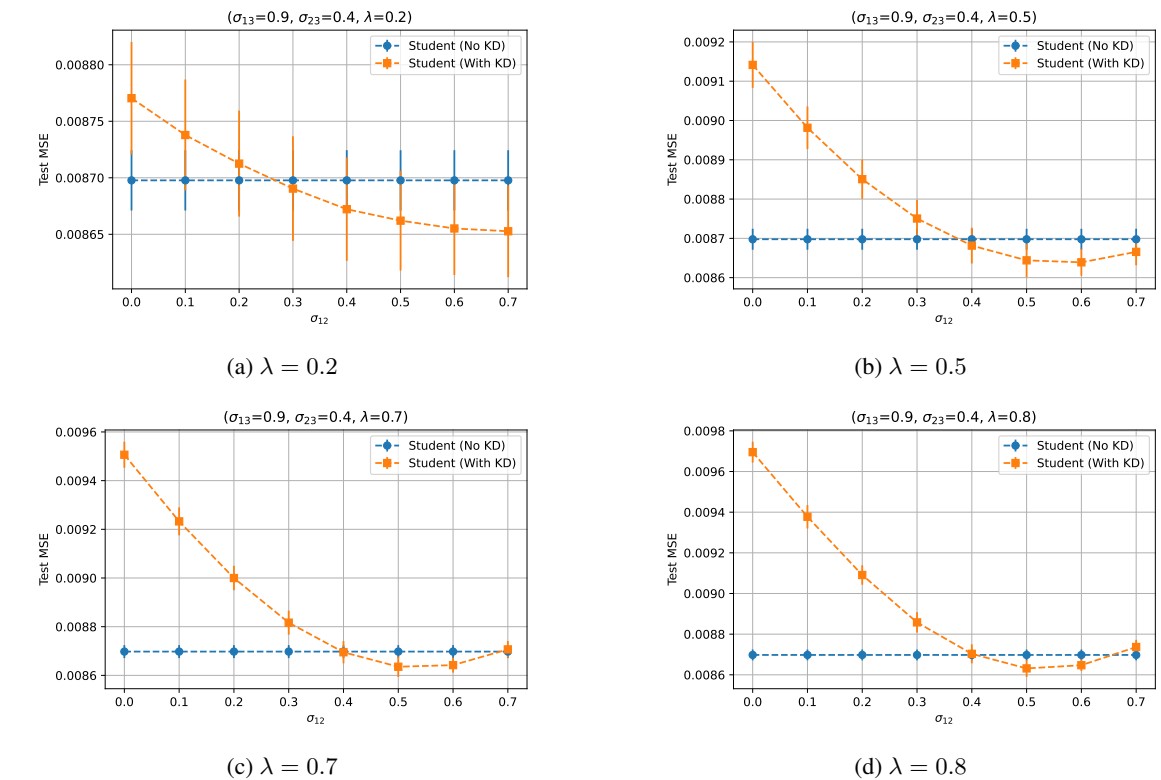

Figure 4: Test MSE on synthetic regression data for varying distillation weight $\lambda$. Orange dashed curves: student with KD; blue dashed curves: student without KD.

Figure 4 reports test mean-squared error (MSE) as a function of the inter-modality correlation $\sigma_{12}$ for distillation weights $\lambda \in \{0.2, 0.5, 0.7, 0.8\}$. Because varying only $\lambda$ does not change the learned representations' mutual information (MI), the MI curves coincide with those obtained at $\lambda = 0.3$ (see Fig. 1). From Fig. 4, when $\sigma_{12}$ is large (e.g., $\sigma_{12} = 0.7$, indicating strong teacher–student alignment), distillation improves the student provided two conditions hold: (i) the CCH criterion $I(H_1; H_2) > I(H_2; Y)$ and (ii) a sufficiently small $\lambda$ to avoid over-regularizing toward the teacher. This behavior is consistent with Theorem 1.

Table 9: Network architecture for synthetic experiments.

| Layer | # Units | Activation |
|---|---|---|
| Input | 100 | – |
| Linear | 64 | ReLU |
| Linear | 1 | – |

*To directly address the more realistic setting where the teacher has higher capacity than the student, we have now performed an additional synthetic experiment in which the teacher network is strictly larger than the student.*

Table 10: Training configuration and dataset details for synthetic experiments.

| Item | Value |
|------|-------|
| Training dataset | Synthetic Gaussian |
| Train/Test split | 10,000 / 5,000 |
| Optimizer | Adam |
| Learning rate | 0.01 |
| Epochs | 300 |

*The detailed architectures of the teacher and student networks are reported in Tables 11 and 12, respectively, and the training configuration is summarized in Table 13. As shown, the teacher is a wider multilayer perceptron with hidden layers of sizes $128$ and $64$, while the student uses significantly smaller hidden layers of sizes $32$ and $16$.*

Table 11: Teacher network architecture for synthetic experiments.

| Layer | Units | Activation |
|-------|-------|------------|
| Input layer | 100 | – |
| Linear layer | 128 | ReLU |
| Linear layer | 64 | ReLU |
| Linear layer | 1 | – |

Table 12: Student network architecture for synthetic experiments.

| Layer | Units | Activation |
|-------|-------|------------|
| Input layer | 100 | – |
| Linear layer | 32 | ReLU |
| Linear layer | 16 | ReLU |
| Linear layer | 1 | – |

*The results are reported in Figure 5. Panel 5a shows the student test MSE with and without KD as a function of the teacher–student correlation $\sigma_{12}$, while Panel 5b reports the corresponding mutual information between representations, $I(H_1; H_2)$ and $I(H_2; Y)$. Consistent with our Cross-modal Complementarity Hypothesis (CCH), we again observe that KD is beneficial precisely in the regime where $I(H_1; H_2) > I(H_2; Y)$. Moreover, comparing these results with the equal-architecture setting, we find that when the teacher has higher capacity than the student, the performance gains from KD are* larger*: the KD-trained student achieves a more pronounced reduction in test MSE relative to its no-KD counterpart. This indicates that our CCH-based criterion continues to predict KD effectiveness even when teacher and student have different capacities, and that a higher-capacity teacher can further amplify the benefits of cross-modal distillation rather than being an artifact of using identical architectures.*

## C  EXPERIMENTAL DETAILS AND RESULTS FOR IMAGE DATA

We evaluate our approach using the MNIST (LeCun et al., 1998) and MNIST-M (Ganin and Lempitsky, 2015) datasets. MNIST comprises 70,000 $28 \times 28$ grayscale images of handwritten digits (0–9). MNIST-M adapts these digits by blending them onto natural-image backgrounds sampled from the BSDS500 dataset (Martin et al., 2001), resulting in colored images with identical labels (Figure 6). Below, we detail the MNIST-M construction, the network architecture, training configuration, and additional results for varying blending coefficients.

Table 13: Training configuration and dataset details for synthetic experiments.

| Training Parameter | Value |
|---|---|
| Dataset | Synthetic data |
| Train/Test split | 10,000 / 5,000 |
| Optimizer | Adam |
| Learning rate | 0.02 |
| Epochs | 300 |

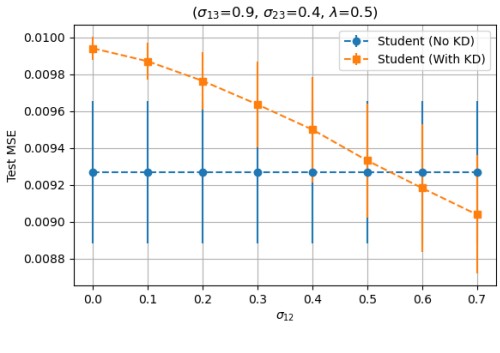

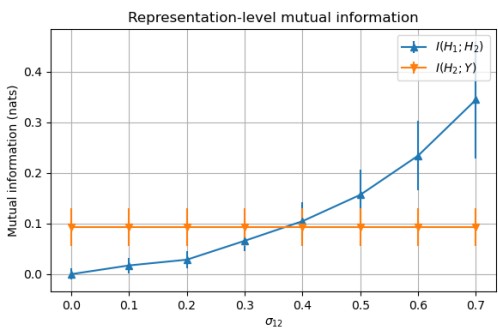

(a) Student MSE vs. $\sigma_{12}$.

(b) Representation MI vs. $\sigma_{12}$.

Figure 5: Regression results on synthetic data when the architectures of teacher network is larger than that of student network. Results demonstrate that when the correlation between teacher modality and student modality ($\sigma_{13}$) surpasses the correlation between student modality and label ($\sigma_{23} = 0.4$), the student network trained with KD achieves consistently lower test MSE compared to training without KD.

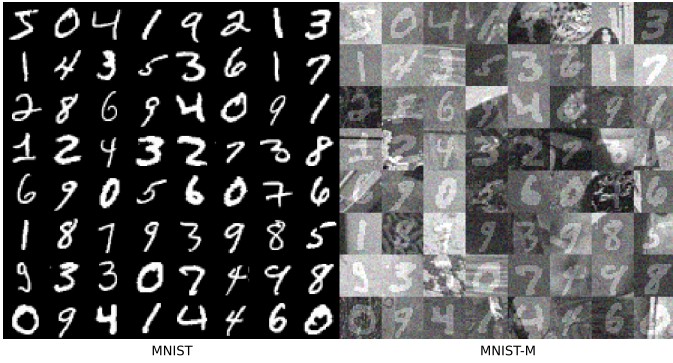

Figure 6: Sample images from MNIST (left) and MNIST-M (right).

---

**Algorithm 1:** Cross-modal knowledge distillation protocol for image data

---

**Input:** MNIST and MNIST-M datasets
**Output:** Test accuracy of student with and without distillation
1: **Teacher pretraining:** Train a teacher network on MNIST;
2: **Student baseline:** Train a student network on MNIST-M using only ground-truth labels;
3: **Distillation:**;
4:  Freeze teacher parameters;
5:  **for** *each Gaussian blur level* $\gamma$ **do**
6:    Apply Gaussian blur of intensity $\gamma$ to teacher inputs;
7:    Obtain soft targets from the frozen teacher;
8:    Train a new student on MNIST-M using both labels and soft targets (Eq. 2);
9: **Evaluation:** Evaluate both student models on the MNIST-M test set;

---

To generate each MNIST-M image, we first binarize the original MNIST digit via thresholding and replicate the resulting single-channel image across the red, green, and blue channels, ensuring compatibility with RGB-based network architectures while preserving the digit's grayscale silhouette. We apply a luminance-preserving transformation to convert BSDS500 patches to grayscale, matching the teacher modality. We then extract a random $28 \times 28$ patch $I_{\mathrm{BSDS}}$ from the processed BSDS500 images and compute:

$$I_{\mathrm{MNISTM}} = \alpha\, I_{\mathrm{MNIST}} + (1-\alpha)\, I_{\mathrm{BSDS}},$$

where $\alpha \in [0,1]$ controls the digit's prominence over the background. Having specified the MNIST-M construction, we conduct training and evaluation according to Algorithm 1. For the experiments in Figure 2 and Table 1, we set $\alpha = 0.2$.

Both teacher and student models share the architecture listed in Table 14 and the training parameters in Table 15. We train using stochastic gradient descent (learning rate 0.002, 100 epochs) with a distillation temperature of $T = 3$ and a loss weight $\lambda = 0.5$. All experiments were executed on an NVIDIA A100 GPU.

Table 14: Network architecture for image experiments.

| Operation | Size | Activation |
|---|---|---|
| Input $\rightarrow$ Linear layer | 1024 | LeakyReLU |
| Linear layer | 256 | LeakyReLU |
| Linear layer | 10 | – |

Table 15: Training configuration and dataset details for image experiments.

| Training Dataset | MNIST / MNIST-M |
|---|---|
| Train/Test Split | 60000 / 10000 |
| Optimizer | SGD |
| Learning Rate | 0.002 |
| Epochs | 100 |
| $T$ | 3 |
| $\lambda$ | 0.5 |

Table 16 presents results for $\alpha = 0.18$ under the same settings. First, the sign of the student accuracy difference (Student Acc Diff) precisely matches that of the mutual-information gap (MI GAP), thereby confirming the CCH. Second, compared to the $\alpha = 0.2$ setting shown in Figure 2, the lower blending weight

reduces the mutual information shared between the MNIST (teacher) and MNIST-M (student) modalities. This reduction in shared information corresponds to a diminished—sometimes negative—distillation gain, demonstrating that student performance declines as the teacher–student mutual information decreases.

Table 16: Experimental results for the MNIST/MNIST-M dataset for $\alpha = 0.18$. MNIST is the teacher modality and MNIST-M is the student modality. The teacher network achieves a test accuracy score of $0.9812 \pm 0.0003$ and $I(H_{\text{teacher}}; Y) = 1.9095$.

| Gamma Level | $I(H_{\text{teacher}}; H_{\text{student}})$ | $I(H_{\text{student}}; Y)$ | Student KD Acc | Student No-KD Acc | MI GAP | Student Acc Diff |
|---|---|---|---|---|---|---|
| 0 | 1.3956 | 1.2685 | $0.8484 \pm 0.0019$ | $0.8338 \pm 0.0034$ | 0.1271 | $0.0146 \pm 0.0052$ |
| 0.5 | 1.2949 | 1.2685 | $0.8425 \pm 0.0042$ | $0.8338 \pm 0.0034$ | 0.0264 | $0.0087 \pm 0.0070$ |
| 1.5 | 1.2533 | 1.2685 | $0.8296 \pm 0.0017$ | $0.8338 \pm 0.0034$ | -0.0152 | $-0.0042 \pm 0.0034$ |
| 2.5 | 0.9472 | 1.2685 | $0.6216 \pm 0.0243$ | $0.8338 \pm 0.0034$ | -0.3213 | $-0.2122 \pm 0.0232$ |
| 3.5 | 0.7817 | 1.2685 | $0.3325 \pm 0.0179$ | $0.8338 \pm 0.0034$ | -0.4868 | $-0.5013 \pm 0.0190$ |

## D  EXPERIMENTAL DETAILS FOR CMU-MOSEI DATA

The CMU Multimodal Opinion Sentiment and Emotion Intensity (CMU-MOSEI) dataset contains 23,453 video segments annotated for sentiment and emotion. Each segment includes time-aligned transcriptions, audio, and visual data, providing three distinct modalities. Our preprocessing protocol for these modalities is detailed in the Algorithm 2.

---
**Algorithm 2:** MOSEI Preprocessing Protocol

---
**Input :** CMU-MOSEI utterance-level dataset: text; time-aligned audio & visual feature streams.
**Data & splits:** Use the official train/validation/test partition.
**Text:** Tokenize texts and map tokens to pretrained word embeddings; treat *one token = one timestep*.
**foreach** *utterance $u$ in the dataset* **do**
    **Temporal alignment:** Find the first non-padding token index $s$ in text($u$); slice *text/audio/vision* to start at $s$ (text defines the time base).
    **Length control:** For each modality, truncate to at most $L=50$ steps, then right-pad with zeros to exactly $L$.
**Labels:** For classification, set $y=1$ if sentiment score $>0$, else $y=0$.
**Batching:** Collate as *(vision, audio, text, label)* to form shapes $(B, L, D_v)$, $(B, L, D_a)$, $(B, L, D_t)$; labels $(B, 1)$; here $D_v = 713$, $D_a = 74$ and $D_t = 300$.

---

The network architecture is identical for all three modalities and is specified in Table 17. The architecture includes a temporal mean-pooling layer, which operates as follows: for a given batch of sequences $X \in \mathbb{R}^{B \times L \times D}$, the layer averages feature vectors across the time dimension $L$ to produce an output $Z \in \mathbb{R}^{B \times D}$, where:

$$Z_{b,d} = \frac{1}{L} \sum_{l=1}^{L} X_{b,l,d} \qquad (b = 1, \ldots, B; \ d = 1, \ldots, D).$$

The training configuration details are consistent across all models and are summarized in Table 18.

## E  EXPERIMENTAL DETAILS AND RESULTS FOR CANCER DATA

For cancer data, Table 19 summarizes the subtype distributions. For the experiments of Tables 5–7, the teacher and student networks share the same architecture used in the synthetic data experiments (see Table 9). Table 20 summarizes the training configurations and dataset splits for the three cancer cohorts.

Table 17: Network architecture for the CMU-MOSEI experiments.

| Operation | Size | Activation |
|---|---|---|
| Input $(B{\times}L{\times}D) \rightarrow$ Temporal Mean-Pool $\rightarrow$ Flatten | $B{\times}L{\times}D \rightarrow B{\times}D$ | – |
| Linear Layer | $D \rightarrow 256$ | ReLU |
| BatchNorm1d + Dropout ($p{=}0.3$) | – | – |
| Linear Layer | $256 \rightarrow 128$ | ReLU |
| BatchNorm1d + Dropout ($p{=}0.3$) | – | – |
| Linear Layer (Classifier Head) | $128 \rightarrow 2$ | – |

Table 18: Training configuration and dataset details for CMU-MOSEI experiments.

| Training Dataset | CMU-MOSEI |
|---|---|
| Train/Validation/Test Split | 70% / 10% / 20% |
| Optimizer | AdamW |
| Learning Rate | 0.0005 |
| LR Schedule | CosineAnnealingLR ($T_{\max} =$ epochs, $\eta_{\min} = 0$) |
| Epochs | 100 |
| Temperature ($T$) | 4.5 |
| Distillation Weight ($\lambda$) | 0.5 |

We evaluated two multimodal fusion strategies: direct fusion and fusion with knowledge distillation (Fusion + KD) (Table 8). Both strategies adopt the architecture in Table 21, which uses separate encoders for each modality followed by feature concatenation (see Figure 3); each encoder comprises 64 units. In the cross-modal distillation protocol (Tables 5–7), we pretrained the teacher network on its modality and then used its soft targets to guide the student (Algorithm 1). By contrast, the fusion experiments train both encoders jointly—without teacher pretraining—while applying a distillation loss to transfer knowledge. Table 22 lists the corresponding training parameters.

To demonstrate the generality of our approach beyond the KIPAN cohort, we also conducted experiments on BRCA data. Table 23 reports the performance metrics for direct fusion and Fusion + KD, and Table 24 lists the corresponding training settings. Across all teacher–student pairs, the mutual information between teacher and student representations consistently exceeds that between student representations and labels, and the Fusion+KD strategy outperforms direct fusion, thereby corroborating the CCH.

*We further consider a three-modality fusion setting that jointly uses mRNA, RPPA, and CNV. On BRCA, mRNA exhibits the highest mutual information with the label, whereas CNV has the lowest. Guided by the CCH, we therefore apply KD only from mRNA to CNV (treating mRNA as the teacher and CNV as the student), while all three modalities are fused at prediction time. The resulting performance is reported in Table 25. These reults are also align with the CCH.*

## F  METHODS FOR MUTUAL INFORMATION ESTIMATION

Mutual information quantifies the dependency between random variables, but its estimation remains challenging, especially when the underlying probability distributions are unknown. Exact mutual information computation is tractable only for small datasets with known distributions. To address this limitation, Kraskov et al. (2004) introduced a k-nearest neighbors (kNN) estimator for mutual information between continuous random variables. This estimator was further extended by Ross (2014) to handle cases where one variable is discrete and the other continuous—a critical adaptation given that many real-world datasets involve mixed

Table 19: Subtype distribution for the BRCA, KIPAN, and LIHC cohorts.

|  | **BRCA** | **KIPAN** | **LIHC** |
|---|---|---|---|
| **Subtypes** | Normal-like: 44
Basal-like: 129
HER2-enriched: 49
Luminal A: 338
Luminal B: 267 | KICH: 63
KIRC: 492
KIRP: 212 | Blast-Like: 39
CHOL-Like: 18
Liver-Like: 113 |

Table 20: Training configuration and dataset details for cross-modal distillation experiments on BRCA, LIHC cancer data.

| **Training Dataset** | BRCA, LIHC |
|---|---|
| Train/Test Split | 90% / 10% |
| Optimizer | Adam |
| Learning Rate | 0.01 |
| Epochs | 200 |
| Temperature ($T$) | 2 |
| Distillation Weight ($\lambda$) | 0.5 |

data types. More recent approaches, such as Mutual Information Neural Estimation (MINE) (Belghazi et al., 2018), leverage neural networks to estimate mutual information between high-dimensional continuous variables. Additionally, a novel method known as latent mututal information (LMI) has been developed (Gowri et al., 2024), which applies a nonparametric mutual information estimator to low-dimensional representations extracted by a theoretically motivated model architecture.

Table 21: Layer-by-layer specification for multimodal fusion experiments on cancer data.

| Branch | Layer | I/O | Act. | Notes |
|---|---|---|---|---|
| **Modality 1** | Linear | $n_{\text{inputMod1}} \rightarrow n_{\text{enc}}$ | ReLU | FC projection |
|  | BatchNorm1d | $n_{\text{enc}} \rightarrow n_{\text{enc}}$ | — | Normalization |
|  | Dropout | $n_{\text{enc}}$ | — | $p = 0.25$ |
| **Modality 2** | Linear | $n_{\text{inputMod2}} \rightarrow n_{\text{enc}}$ | ReLU | FC projection |
|  | BatchNorm1d | $n_{\text{enc}} \rightarrow n_{\text{enc}}$ | — | Normalization |
|  | Dropout | $n_{\text{enc}}$ | — | $p = 0.25$ |
| **Fusion & Classification** | Concat | $2\,n_{\text{enc}}$ | — | Merge embeddings |
|  | Linear (fusion) | $2\,n_{\text{enc}} \rightarrow n_{\text{classes}}$ | — | Joint-feature logits |
|  | Linear (modality) | $n_{\text{enc}} \rightarrow n_{\text{classes}}$ | — | Modality-specific logits |

Table 22: Training configuration and dataset details for multimodal fusion experiments on KIPAN data.

| Training Dataset | KIPAN |
|---|---|
| Train/Test Split | 90% / 10% |
| Optimizer | Adam |
| Learning Rate | 0.007 |
| Epochs | 200 |
| Temperature ($T$) | 1 |
| Distillation Weight ($\lambda$) | 0.5 |

Table 23: Overall multimodal performance of direct fusion and Fusion+KD on BRCA, reported with mutual information of modality representations (teacher–label, teacher–student, student–label).

| | Mutual Information | | | Fusion | | | | Fusion+KD | | | |
|---|---|---|---|---|---|---|---|---|---|---|---|
| | Teacher–Label | Teacher–Student | Student–Label | Acc | AUC | Macro F1 | Weighted F1 | Acc | AUC | Macro F1 | Weighted F1 |
| mRNA (teacher) CNV (student) | 1.1081 | 0.5057 | 0.2757 | 0.7711 | 0.9157 | 0.6432 | 0.7563 | 0.8434 | 0.8610 | 0.6533 | 0.8225 |
| RPPA (teacher) CNV (student) | 0.7328 | 0.3367 | 0.2757 | 0.5663 | 0.7844 | 0.5604 | 0.5715 | 0.6024 | 0.7929 | 0.5897 | 0.6103 |

Table 24: Training configuration and dataset details for multimodal fusion experiments on BRCA.

| Training Dataset | MNIST / MNIST-M |
|---|---|
| Train/Test Split | 90% / 10% |
| Optimizer | Adam |
| Learning Rate | 0.04 |
| Epochs | 200 |
| Temperature ($T$) | 4 |
| Distillation Weight ($\lambda$) | 0.5 |

Table 25: Performance of direct fusion and Fusion+KD on BRCA when fusing three modalities (mRNA, CNV, and RPPA). KD is applied only from mRNA (teacher) to CNV (student).

| | Fusion | | | | Fusion+KD | | | |
|---|---|---|---|---|---|---|---|---|
| | Acc | AUC | Macro F1 | Weighted F1 | Acc | AUC | Macro F1 | Weighted F1 |
| mRNA (teacher) CNV (student) RPPA | 0.855 | 0.861 | 0.668 | 0.83 | 0.868 | 0.875 | 0.673 | 0.832 |

