# OpenReview forum: "Information-Theoretic Criteria for Knowledge Distillation in Multimodal Learning"
_ICLR.cc/2026/Conference — Submitted to ICLR 2026_

### Official Review · Reviewer_nSot · 2025-10-26

**Soundness:** 2
**Presentation:** 2
**Contribution:** 2
**Rating:** 4
**Confidence:** 2

**Summary:**

This paper introduces the Cross-modal Complementarity Hypothesis (CCH), a theoretical framework for understanding when cross-modal knowledge distillation (KD) improves performance in multimodal learning. The hypothesis posits that KD is effective when the mutual information between teacher and student representations exceeds that between student representations and the labels. The authors validate CCH both theoretically—via a joint Gaussian model—and empirically across diverse datasets, including synthetic data, MNIST/MNIST-M, CMU-MOSEI, and cancer-related omics datasets. They demonstrate that CCH reliably predicts when KD will enhance student performance and offer practical guidance for selecting teacher modalities.

**Strengths:**

- The paper is generally well-written and well-motivated.
- The organisation of the paper is clear.

**Weaknesses:**

- MFH already posits that the success of cross-modal KD depends on the degree of label-relevant information present in the teacher modality and its alignment with the student. CCH formalizes this intuition using mutual information inequalities, but the underlying insight appears to be a refinement rather than a fundamentally new idea.

- All experiments use identical architectures for teacher and student models to isolate mutual information effects. This setup may not reflect realistic KD scenarios where teacher and student can differ in capacity and architecture.

- Its a bit confusing to use MNIST and MNIST-M as the teacher and student modality, as they are basically the same modality but from different domain. I think the author should use AV-MNIST instead.

- its a bit confusing to see $I(H_{text}; H_{audio})$. Please specifcy how is this computed.

**Questions:**

n/a

---

> ### Author Response · Authors · 2025-11-24
>
> **1. Weakness 1: MFH already posits that the success of cross-modal KD depends on the degree of label-relevant information present in the teacher modality and its alignment with the student. CCH formalizes this intuition using mutual information inequalities, but the underlying insight appears to be a refinement rather than a fundamentally new idea.**
>
> Thank you for pointing this out. MFH provides a qualitative intuition—that KD depends on how much label-relevant information the teacher contains—but it does not offer a computable or verifiable condition. In contrast, CCH turns this intuition into a quantitative and testable criterion based on mutual information that can be evaluated directly from representations. Our experiments further show that this criterion reliably predicts when KD succeeds or fails, providing practical value beyond the qualitative insight in MFH.
>
> **2. Weakness 2: All experiments use identical architectures for teacher and student models to isolate mutual information effects. This setup may not reflect realistic KD scenarios where teacher and student can differ in capacity and architecture.**
>
> Thank you for raising this concern. Our choice of using identical architectures for the teacher and student networks was intentional, as it allowed us to isolate the effect of mutual information without conflating it with capacity differences. To directly address the more realistic setting where the teacher has higher capacity than the student, **we have now added an additional synthetic experiment** in which the teacher network is strictly larger than the student, reported **in Appendix B**.
>
> **3. Weakness 3: Its a bit confusing to use MNIST and MNIST-M as the teacher and student modality, as they are basically the same modality but from different domain. I think the author should use AV-MNIST instead.**
>
> Thank you for raising this concern. MNIST-M is created by compositing MNIST digits onto natural-image backgrounds, which yields inputs that differ substantially in low-level statistics while sharing the same semantic label space. In our study, we deliberately adopt this closely related pair as a controlled teacher–student setting: because the architectures and labels are matched and the “modality gap” is modest, changes in student performance can be more directly attributed to variations in the effective information shared between teacher and student, rather than to confounding factors.
>
> We agree that benchmarks like AV-MNIST, which couple genuinely heterogeneous audio and visual streams, are also natural testbeds for cross-modal distillation. However, such setups introduce additional sources of variability (e.g., audio-specific optimization issues) that can obscure the specific role of mutual information in driving successful knowledge transfer. In our experimental design, this “toy but controlled” MNIST/MNIST-M pair is complemented by CMU-MOSEI and TCGA, where we investigate the same Cross-modal Complementarity Hypothesis in realistic multimodal scenarios involving text, vision, audio, and omics data. Taken together, these experiments allow us to both isolate the core phenomenon in a near-ideal setting and demonstrate that the observed CCH behavior extends to truly heterogeneous, real-world multimodal tasks.
>
> **4. Weakness 4: ts a bit confusing to see $I(H_{text},H_{audio})$. Please specifcy how is this computed.**
>
> Thank you for pointing this out. We apologize for the lack of clarity.
>
> In our notation, $H_{\text{text}}$ and $H_{\text{audio}}$ denote the **learned representations** of the text and audio modalities, not the raw inputs. Concretely, in the CMU-MOSEI experiments each modality is passed through its own encoder. After temporal mean pooling and the two fully connected layers, we obtain a fixed-dimensional vector for each utterance; we denote these vectors by
> $h_{\text{text}}^{(i)} \in \mathbb{R}^{128}$,
> $h_{\text{audio}}^{(i)} \in \mathbb{R}^{128},$
> for the $i$-th utterance.
>
> The mutual information $I(H_{\text{text}}; H_{\text{audio}})$ is estimated from the paired samples
> $
> \{(h_{\text{text}}^{(i)}, h_{\text{audio}}^{(i)})\}_{i=1}^n
> $
> using the latent mutual information (latentmi) estimator, as described in Appendix F.

---

### Official Review · Reviewer_UHPd · 2025-10-27

**Soundness:** 2
**Presentation:** 3
**Contribution:** 2
**Rating:** 2
**Confidence:** 3

**Summary:**

The paper introduces the Cross-modal Complementarity Hypothesis (CCH), proposing that cross-modal knowledge distillation (KD) is effective when the mutual information between teacher and student representations exceeds that between the student and the labels (I(H₁;H₂) > I(H₂;Y)). The authors support this claim through Gaussian-model analysis and extensive empirical validation across synthetic, vision, audio-visual, and biomedical datasets. The work aims to provide a principled, information-theoretic understanding of when cross-modal distillation helps.
However, despite a systematic set of experiments, the theoretical contribution remains shallow and largely intuitive. The proposed criterion essentially reformulates a well-known intuition that KD is useful only when the teacher carries additional label-relevant information that the student lacks, without introducing new theoretical insights or addressing the practical challenges of mutual information estimation.

**Strengths:**

1. The paper is clearly written and well organized, with a coherent conceptual narrative and a solid empirical structure.
2. It consolidates a range of existing intuitions about when knowledge distillation is effective into a single, measurable information-theoretic framework.
3. The experimental evaluation is diverse and carefully executed, and the accompanying code and datasets are well documented, enhancing empirical transparency and reproducibility.

**Weaknesses:**

1. The central criterion (I(H₁;H₂) > I(H₂;Y)) essentially restates an intuitive idea that distillation is beneficial only when the teacher provides additional label-relevant information that the student lacks. This condition is conceptually simple and offers limited theoretical novelty beyond rephrasing a well-known intuition in information-theoretic terms.
2. The Gaussian analysis functions more as a didactic toy example than as a rigorous theoretical derivation. It does not uncover any new properties of mutual information, nor does it deepen our understanding of how knowledge distillation operates in nonlinear or high-dimensional settings.
3. The hypothesis describes a correlative rather than a causal relationship, offering no explanation of why or how mutual information alignment leads to improved performance. It provides little insight into the internal mechanisms of representation transfer.
4. The empirical evaluation primarily confirms the hypothesis rather than attempting to challenge or falsify it. The observed correlation between MI gap and performance improvement is largely expected and does not establish theoretical necessity.
5. The proposed condition is difficult to estimate reliably in high-dimensional multimodal models, which limits its practicality as a predictive or diagnostic tool for real-world knowledge distillation systems.

**Questions:**

1. Could the authors clarify how the proposed CCH framework fundamentally differs from prior approaches that explicitly maximize teacher–student mutual information, such as Variational Information Distillation (VID) or Contrastive KD?
2. Are there scenarios where the condition I(H₁;H₂) > I(H₂;Y) holds but distillation does not yield improvement? Providing such counterexamples could help delineate the boundaries and limitations of the hypothesis.
3. Can the authors establish a clearer theoretical or intuitive link between mutual information and the optimization dynamics of distillation, for example in terms of gradient alignment or feature transfer behavior?
4. How sensitive are the reported empirical results to the specific mutual information estimator used (e.g., MINE, latentMI, KSG) and to variations in data dimensionality?
5. What would be the practical implications or implementations of this criterion in modern large-scale multimodal transformer architectures, where direct MI estimation is computationally infeasible?

---

> ### Author Response · Authors · 2025-11-24
>
> **1. Weakness 1: The central criterion ($I(H_1,H_2)>I(H_2,Y)$) essentially restates an intuitive idea that distillation is beneficial only when the teacher provides additional label-relevant information that the student lacks. This condition is conceptually simple and offers limited theoretical novelty beyond rephrasing a well-known intuition in information-theoretic terms.**
>
> Thank you for this observation. Our goal is not merely to restate an intuition, but to formalize it in a quantitative, verifiable, and predictive manner. Although the high-level idea—that distillation helps when the teacher contains information the student lacks—may appear intuitive, prior work has not provided a precise, computable criterion which is verified across modalities. Thus, the contribution lies not in proposing an entirely new intuition, but in turning an informal belief into a theoretically grounded, testable rule that consistently predicts KD behavior in practice.
>
> **2. Weakness 2: The Gaussian analysis functions more as a didactic toy example than as a rigorous theoretical derivation. It does not uncover any new properties of mutual information, nor does it deepen our understanding of how knowledge distillation operates in nonlinear or high-dimensional settings.**
>
> We agree that the Gaussian setting is a simplification, and we intentionally use it as a minimal model rather than a full theory of KD. Its purpose is to provide a clean analytical environment where the CCH inequality can be derived exactly.
>
> Our empirical results show that behaviors predicted in the Gaussian model—particularly the thresholding effect of persist across nonlinear networks and real data. Thus, although simplified, the Gaussian model captures structurally correct behavior that generalizes surprisingly well.
>
> In addition, **we have added a paragraph in Section 3** discussing how the criterion extends to non-linear theoretical settings.
>
> **3. Weakness 3: The hypothesis describes a correlative rather than a causal relationship, offering no explanation of why or how mutual information alignment leads to improved performance. It provides little insight into the internal mechanisms of representation transfer.**
>
> We thank the reviewer for raising this point. Contrary to the concern that our hypothesis is purely correlative, our analysis explicitly explains why mutual-information alignment leads to improved performance. In particular, we show that KD is equivalent to training the student on an effective label that is a convex combination of the ground-truth label and the teacher’s prediction. In the Gaussian setting, the resulting risk decomposes into (i) a **bias term**, measuring how much the student’s predictor deviates from the optimal predictor based on its own modality $X_2$, and (ii) a **variance/noise term**, capturing teacher-induced noise that is not representable in the student’s feature space. The inequality $I(H_1;H_2) > I(H_2;Y)$ precisely characterizes the regime where the teacher’s output carries additional label-relevant information that lies in a subspace accessible to the student. In this regime, the KD term increases the signal-to-noise ratio of the effective labels: the added information primarily reduces bias rather than adding irreducible noise, leading to a strict reduction in excess risk. When the inequality is reversed, the teacher’s signal is either redundant or largely unobservable from $H_2$, so the KD gradient behaves more like noise and does not improve (and can even worsen) performance.
>
> **4. Weakness 4: The empirical evaluation primarily confirms the hypothesis rather than attempting to challenge or falsify it. The observed correlation between MI gap and performance improvement is largely expected and does not establish theoretical necessity.**
>
> We appreciate this concern. Our experiments were designed not only to confirm the hypothesis but also to actively probe for counter-examples by systematically varying teacher quality, modality pairs, training budgets, dataset sizes, and MI estimators. In particular, we include settings where KD hurts performance when the MI gap is negative, which would contradict the hypothesis if improvements were observed. Across these stress tests, the empirical evaluation consistently aligns with and thus supports the theoretical prediction.
>
> In addition, we agree that high teacher–student similarity alone might be intuitively associated with better KD. However, our contribution is to show that it is not teacher–student MI alone that matters, but rather its relationship to the student–label MI, and that this MI gap consistently predicts when KD should be used or even avoided across very different domains (vision, language) and MI estimators. While we do not claim a universal necessity result beyond the joint Gaussian setting analyzed in the paper, to our knowledge this precise, operational criterion and its systematic empirical validation have not been established before.

---

> > ### Author Response · Authors · 2025-11-24
> >
> > **5. Weakness 5: The proposed condition is difficult to estimate reliably in high-dimensional multimodal models, which limits its practicality as a predictive or diagnostic tool for real-world knowledge distillation systems.**\
> > **Question 4: How sensitive are the reported empirical results to the specific mutual information estimator used (e.g., MINE, latentMI, KSG) and to variations in data dimensionality?**
> >
> > Thank you for raising this point. We agree that exact mutual information (MI) computation is infeasible in high-dimensional settings, and that naïve estimators can be both expensive and unstable. Our goal, however, is more modest: CCH only requires a coarse, relative comparison between $I(H_1; H_2)$ and $I(H_2; Y)$, not an accurate estimate of their absolute values. To this end, as detailed in Appendix F, we work at the level of learned representations $H_1, H_2$ and use practical MI estimators specifically designed for high-dimensional data.
> >
> > In particular, the latent mutual information (latentmi) estimator first maps data into a low-dimensional latent space and then applies a nonparametric MI estimator there, which substantially reduces both variance and computational cost. For example, on MNIST we can estimate the label MI in under **10 seconds** on a single GPU, which is negligible compared to the time required to train the corresponding models. Thus, in a realistic model-design pipeline, running MI estimation once per teacher-student pair adds only a small overhead while potentially avoiding many unhelpful KD runs.
> >
> > Regarding robustness, Table 2 shows that different estimators (latentmi, MINE, KSG) indeed yield different absolute MI values, but they consistently agree on the **ordering** of the relevant quantities, for instance, whether $I(H_1; H_2)$ is larger or smaller than $I(H_2; Y)$ across modalities. Since CCH depends only on this ordering (i.e., on the sign of the MI gap $I(H_1; H_2) - I(H_2; Y)$), the decision predicted by CCH—whether KD should help or hurt—is invariant across all estimators we tried. This is further supported by our degradation experiments (Tables 4-7), where changes in the sign of the MI gap reliably track the transition between beneficial and harmful KD, regardless of the exact MI values.
> >
> > In summary, while we fully acknowledge the difficulties of high-dimensional MI estimation, the CCH criterion is deliberately formulated so that it only relies on robust, qualitative information (relative ordering), and our empirical study shows that this information is stable across multiple standard estimators and datasets.
> >
> > **6. Question 1: Could the authors clarify how the proposed CCH framework fundamentally differs from prior approaches that explicitly maximize teacher–student mutual information, such as Variational Information Distillation (VID) or Contrastive KD?**
> >
> > Thank you for the question. Methods such as VID or contrastive KD aim to optimize the teacher–student mutual information during training. In contrast, CCH is an a diagnostic tool: it predicts whether KD will be beneficial before performing any distillation. Thus, our goal is fundamentally different—we do not propose a new KD loss, but a practical criterion for deciding when KD is worthwhile.

---

> > > ### Author Response · Authors · 2025-11-24
> > >
> > > **7. Question 2: Are there scenarios where the condition $I(H_1,H_2)>I(H_2,Y)$ holds but distillation does not yield improvement? Providing such counterexamples could help delineate the boundaries and limitations of the hypothesis.**
> > >
> > > Thank you for the insightful question. **A concrete example already appears in our synthetic experiments in Appendix B**. We fix the data distribution (hence the mutual information curves in Fig. 1(b)) and vary the distillation weight $\lambda$. For large inter-modality correlation $\sigma_{12}$, we are in the region where $I(H_1;H_2) > I(H_2;Y)$ (cf. Fig. 1(b)). However, when $\lambda$ is increased to relatively large values (e.g., $\lambda \ge 0.7$), Fig. 4(c–d) shows that the KD-trained student can have *higher* test MSE than the non-distilled student. Intuitively, a very large $\lambda$ over-regularizes the student towards the teacher, forcing it to track the teacher’s residual errors rather than using the teacher as a mild, complementary signal. This gives precisely the kind of scenario the reviewer asks about: the MI condition holds, but a particular (poor) KD configuration does not improve performance.
> > >
> > >
> > > **8. Question 3: Can the authors establish a clearer theoretical or intuitive link between mutual information and the optimization dynamics of distillation, for example in terms of gradient alignment or feature transfer behavior?**
> > >
> > > Thank you for the question. Our work focuses on providing a practical criterion for predicting when KD is beneficial, rather than analyzing the optimization dynamics of distillation. We agree that understanding how mutual information relates to gradient alignment or feature transfer is an important direction, and we view this as complementary future work beyond the scope of our current contribution.
> > >
> > > **9. Question 5: What would be the practical implications or implementations of this criterion in modern large-scale multimodal transformer architectures, where direct MI estimation is computationally infeasible?**
> > >
> > > Thank you for the question. We agree that exact MI estimation on large multimodal transformers is infeasible. However, the CCH criterion only relies on the relative ordering of two MI terms, which can be estimated efficiently using practical approximations. In practice, one can (i) use smaller surrogate models, (ii) compute MI on low-dimensional projections of representations, or (iii) replace MI with inexpensive proxies such as CCA. These approximations are sufficient to determine the ordering and make the criterion applicable in large-scale settings.

---

### Official Review · Reviewer_akzG · 2025-10-31

**Soundness:** 3
**Presentation:** 3
**Contribution:** 3
**Rating:** 4
**Confidence:** 3

**Summary:**

The paper proposes the \emph{Cross-modal Complementarity Hypothesis} (CCH), an information-theoretic rule-of-thumb for predicting when cross-modal knowledge distillation will help a student model. The key idea is that distillation is beneficial when the mutual information between teacher and student representations exceeds that between the student representation and the label, offering a purportedly a priori criterion for choosing teacher modalities. The authors provide a sufficiency argument under a linear-Gaussian, small-regularization regime and present empirical evidence across synthetic data and several multimodal domains (e.g., digit variants, sentiment/audio-visual-text, and omics), plus a comparison suggesting the criterion can inform when to add KD on top of fusion.

While conceptually tidy, the theoretical support is narrow: guarantees rely on idealized linear assumptions and asymptotics, leaving unclear applicability to nonlinear deep networks and finite-sample training typical in CVPR settings. The “a priori” test hinges on estimating mutual information between learned representations, which itself requires trained models and high-dimensional MI estimators with nontrivial bias/variance---the paper does not convincingly quantify estimation reliability or decision robustness. Empirically, most benchmarks are relatively forgiving; there is little evidence on harder, vision-centric tasks (e.g., dense prediction, detection, long-tail distributions, or noisy/weak teachers) where the hypothesis would be stress-tested. Practical guidance is also limited: the work says \emph{when} to distill but gives scant insight into \emph{how} (temperature/$\lambda$, layer choice, calibration, or resilience to teacher errors). Finally, the connection between KD gains and shared label-relevant information is not entirely new; without stronger theory beyond the linear case or sharper ablations ruling out confounds (capacity, regularization side effects), the contribution feels incremental. Overall, a clean unifying lens with promising intuition, but presently short of a reliable decision rule for modern multimodal vision systems.

**Strengths:**

It positions the hypothesis against related work and argues that, despite prior intuitions, no earlier paper stated a concrete MI-based feasibility condition for cross-modal KD.

**Weaknesses:**

The only formal support is in an idealized linear regression setting under a jointly Gaussian data model with a quadratic KD penalty—far from modern non-linear deep nets used in vision.

The paper must average over 50 runs and shows that absolute MI values vary by estimator, indicating sensitivity that is not quantified with uncertainty or calibration analyses.

The condition relies on mutual information between learned representations H1, H2, which themselves require training to obtain. This blunts the promise of a pre-training decision rule.

The MI toolbox (KSG, MINE, latent-MI) is itself known to be challenging in high-D; the paper’s overview underscores that exact MI is intractable beyond small, known-distribution cases, raising questions about robustness as a gating signal.

**Questions:**

How sensitive is the CCH decision to finite-sample noise? Can you bound the probability of a wrong decision (sign flip) as a function of sample size and MI-estimator error?

---

> ### Author Response · Authors · 2025-11-24
>
> **1. Weakness 1: The only formal support is in an idealized linear regression setting under a jointly Gaussian data model with a quadratic KD penalty—far from modern non-linear deep nets used in vision.**
>
> Thank you for raising this point. While our theoretical analysis indeed relies on joint Gaussian assumptions, **we have added a paragraph in Section 3** discussing how the criterion extends to non-linear theoretical settings. We have also conducted extensive empirical studies demonstrating that our criterion remains accurate far beyond this idealized setting. In Section 4.1, we show that the criterion continues to hold under nonlinear models. Moreover, Sections 4.2–4.4 validate the criterion on real-world image, video, text, and cancer datasets, which naturally involve non Gaussianity, complex nonlinearities, and various practical asymmetries. These results suggest that the sufficiency conditions are robust in substantially more realistic scenarios than those covered by the theory.
>
> **2. Weakness 2: The paper must average over 50 runs and shows that absolute MI values vary by estimator, indicating sensitivity that is not quantified with uncertainty or calibration analyses.**
>
> Thank you for raising this point. We averaged over 50 runs only to estimate the standard error of each MI estimator. As shown in Table 1, the variance is significantly smaller than the MI gap relevant to the CCH criterion, meaning the decision is highly stable. In practice, a single run is sufficient, and the criterion is not sensitive to estimator variability at the scale that affects the conclusion.
>
> **3. Weakness 3: The condition relies on mutual information between learned representations H1, H2, which themselves require training to obtain. This blunts the promise of a pre-training decision rule.**
>
> We thank the reviewer for pointing out this potential limitation. Our claim is not that the CCH can be evaluated before any model training whatsoever, but that it provides a pre–distillation (rather than fully post-hoc) decision rule for whether cross-modal KD is likely to help.
>
> In all our experiments, the representations $H_1, H_2$ correspond to:
> (i) a teacher trained on modality $X_1$, and
> (ii) a baseline student trained on modality $X_2$.
> These models must be trained anyway to (a) obtain a strong teacher and (b) get a unimodal student baseline, which are standard steps in cross-modal KD setups. The CCH then only requires measuring mutual information on these existing encoders, which adds negligible computational overhead beyond usual practice.
>
> **4. Weakness 4: The MI toolbox (KSG, MINE, latent-MI) is itself known to be challenging in high-D; the paper’s overview underscores that exact MI is intractable beyond small, known-distribution cases, raising questions about robustness as a gating signal.**
>
> Thank you for raising this point. We agree that exact mutual information (MI) computation is infeasible in high-dimensional settings, and that naïve estimators can be both expensive and unstable. Our goal, however, is more modest: CCH only requires a coarse, relative comparison between $I(H_1; H_2)$ and $I(H_2; Y)$, not an accurate estimate of their absolute values. To this end, as detailed in Appendix F, we work at the level of learned representations $H_1, H_2$ and use practical MI estimators specifically designed for high-dimensional data.
>
> In particular, the latent mutual information (latentmi) estimator first maps data into a low-dimensional latent space and then applies a nonparametric MI estimator there, which substantially reduces both variance and computational cost. For example, on MNIST we can estimate the label MI in under **10 seconds** on a single GPU, which is negligible compared to the time required to train the corresponding models. Thus, in a realistic model-design pipeline, running MI estimation once per teacher-student pair adds only a small overhead while potentially avoiding many unhelpful KD runs.
>
> Regarding robustness, Table 2 shows that different estimators (latentmi, MINE, KSG) indeed yield different absolute MI values, but they consistently agree on the **ordering** of the relevant quantities, for instance, whether $I(H_1; H_2)$ is larger or smaller than $I(H_2; Y)$ across modalities. Since CCH depends only on this ordering (i.e., on the sign of the MI gap $I(H_1; H_2) - I(H_2; Y)$), the decision predicted by CCH—whether KD should help or hurt—is invariant across all estimators we tried. This is further supported by our degradation experiments (Tables 4-7), where changes in the sign of the MI gap reliably track the transition between beneficial and harmful KD, regardless of the exact MI values.

---

> > ### Author Response · Authors · 2025-11-24
> >
> > **5. Question 1: How sensitive is the CCH decision to finite-sample noise? Can you bound the probability of a wrong decision (sign flip) as a function of sample size and MI-estimator error?**
> >
> > Thank you for raising this point. A key observation is that both $I(H_1,H_2)$ and $I(H_2,Y)$ can be estimated by averaging over the entire training set, and for most commonly used MI estimators the estimation error has the standard rate $O(\delta^2/n)$, where $n$ is the sample size and $\sigma^2$ denotes the estimator variance.
> >
> > Let $\Delta:=|I(H_1,H_2)-I(H_2,Y)|$. A sign flip occurs only when the estimation error exceeds $\Delta$. By concentration bounds (e.g., Chebyshev or Bernstein-type inequalities), the probability of a wrong decision decreases rapidly once
> > $n\gg\frac{\delta^2}{\Delta^2}$.
> >
> > Thus, as long as the two MI values differ by a non-negligible margin, the CCH decision remains stable even under finite-sample noise. This is consistent with our empirical results, where the ordering between the two MI terms is highly robust across datasets and estimators.

---

### Official Review · Reviewer_k5cd · 2025-11-01

**Soundness:** 3
**Presentation:** 3
**Contribution:** 3
**Rating:** 6
**Confidence:** 2

**Summary:**

This work proposes a mutual information–based quantifiable criterion to determine whether cross-modal knowledge distillation can effectively improve the performance of the student model. It reveals that when the mutual information between the teacher and student modalities exceeds that between the student modality and the labels, cross-modal knowledge distillation is beneficial to the student’s performance. Under the joint Gaussian assumption, the paper theoretically proves the validity conditions of the CCH criterion and derives how knowledge distillation contributes to reducing the student model’s risk. Extensive experiments across multiple benchmarks further validate the effectiveness of the proposed strategy.

**Strengths:**

1. Analyzing cross-modal knowledge distillation from the perspective of mutual information is an interesting and insightful attempt.

2. The overall writing is clear and easy to follow.

3. The effectiveness of the CCH criterion is validated across diverse datasets, including synthetic data, image datasets (MNIST/MNIST-M), multimodal sentiment analysis (CMU-MOSEI), and cancer multi-omics data (TCGA).

**Weaknesses:**

1. The CCH criterion relies on estimates of mutual information (MI). However, accurately estimating MI in high-dimensional settings is a well-known challenge and can be computationally expensive. In practical scenarios, particularly during the model design phase, performing an additional and potentially unstable MI estimation to “a priori” decide whether KD will be beneficial may be impractical. This limitation weakens the usefulness of CCH as a true a priori criterion. The paper employs multiple MI estimators (such as latentMI, MINE, and KSG), but the differing results suggest limited robustness.

2. Theorem 1 and its proof are based on very strong assumptions, namely that the data follow a joint Gaussian distribution and both teacher and student models are linear. Although these assumptions help simplify the theoretical analysis, modern deep learning involves highly nonlinear models and complex, non-Gaussian data distributions. It remains unclear to what extent the theoretical guarantees can generalize to real-world cross-modal KD scenarios that rely on deep neural networks.

3. In the classical knowledge distillation setting, the teacher model is typically larger and stronger, while the student is smaller and more efficient. Even if the CCH condition is satisfied, a student with limited capacity may not be able to absorb the complementary information provided by the teacher. It would be helpful for the paper to discuss or empirically analyze how the capacity gap affects the practical value of the CCH criterion.

4. The current work focuses mainly on distillation between two modalities. It would be interesting to explore whether the CCH criterion can be extended to multi-modal interactions involving three or more modalities, where pairwise mutual information relationships and higher-order dependencies become more complex. A discussion or preliminary investigation in this direction could further strengthen the contribution.

**Questions:**

Please see the Weaknesses.

---

> ### Author Response · Authors · 2025-11-24
>
> **1. Weakness 1: The CCH criterion relies on estimates of mutual information (MI). However, accurately estimating MI in high-dimensional settings is a well-known challenge and can be computationally expensive. In practical scenarios, particularly during the model design phase, performing an additional and potentially unstable MI estimation to “a priori” decide whether KD will be beneficial may be impractical. This limitation weakens the usefulness of CCH as a true a priori criterion. The paper employs multiple MI estimators (such as c, MINE, and KSG), but the differing results suggest limited robustness.**
>
> Thank you for raising this point. We agree that exact mutual information (MI) computation is infeasible in high-dimensional settings, and that naïve estimators can be both expensive and unstable. Our goal, however, is more modest: CCH only requires a coarse, relative comparison between $I(H_1; H_2)$ and $I(H_2; Y)$, not an accurate estimate of their absolute values. To this end, as detailed in Appendix F, we work at the level of learned representations $H_1, H_2$ and use practical MI estimators specifically designed for high-dimensional data.
>
> In particular, the latent mutual information (latentmi) estimator first maps data into a low-dimensional latent space and then applies a nonparametric MI estimator there, which substantially reduces both variance and computational cost. For example, on MNIST we can estimate the label MI in under **10 seconds** on a single GPU, which is negligible compared to the time required to train the corresponding models.
>
> Regarding robustness, Table 2 shows that different estimators (latentmi, MINE, KSG) indeed yield different absolute MI values, but they consistently agree on the **ordering** of the relevant quantities. Since CCH depends only on this ordering (i.e., on the sign of the MI gap $I(H_1; H_2) - I(H_2; Y)$), the decision predicted by CCH—whether KD should help or hurt—is invariant across all estimators we tried. This is further supported by our degradation experiments (Tables 4-7), where changes in the sign of the MI gap reliably track the transition between beneficial and harmful KD, regardless of the exact MI values.
>
> **2. Weakness 2: Theorem 1 and its proof are based on very strong assumptions, namely that the data follow a joint Gaussian distribution and both teacher and student models are linear. Although these assumptions help simplify the theoretical analysis, modern deep learning involves highly nonlinear models and complex, non-Gaussian data distributions. It remains unclear to what extent the theoretical guarantees can generalize to real-world cross-modal KD scenarios that rely on deep neural networks.**
>
> Thank you for raising this point. While our theoretical analysis indeed relies on joint Gaussian assumptions, **we have added a paragraph in Section 3** discussing how the criterion extends to non-linear theoretical settings. We have also conducted extensive empirical studies demonstrating that our criterion remains accurate far beyond this idealized setting. In Section 4.1, we show that the criterion continues to hold under nonlinear models. Moreover, Sections 4.2–4.4 validate the criterion on real-world image, video, text, and cancer datasets, which naturally involve non Gaussianity, complex nonlinearities, and various practical asymmetries. These results suggest that the sufficiency conditions are robust in substantially more realistic scenarios than those covered by the theory.
>
> **3. Weakness 3: In the classical knowledge distillation setting, the teacher model is typically larger and stronger, while the student is smaller and more efficient. Even if the CCH condition is satisfied, a student with limited capacity may not be able to absorb the complementary information provided by the teacher. It would be helpful for the paper to discuss or empirically analyze how the capacity gap affects the practical value of the CCH criterion.**
>
> Thank you for raising this concern. Our choice of using identical architectures for the teacher and student networks was intentional, as it allowed us to isolate the effect of mutual information without conflating it with capacity differences. To directly address the more realistic setting where the teacher has higher capacity than the student, **we have now added an additional synthetic experiment** in which the teacher network is strictly larger than the student, reported **in Appendix B**.
>
> **4. Weakness 4: The current work focuses mainly on distillation between two modalities. It would be interesting to explore whether the CCH criterion can be extended to multi-modal interactions involving three or more modalities, where pairwise mutual information relationships and higher-order dependencies become more complex. A discussion or preliminary investigation in this direction could further strengthen the contribution.**
>
> Thank you for raising this point. I have **added three modalities experiments in Section E**.

---

> > ### Comment · Reviewer_k5cd · 2025-11-26
> >
> > Thank you for the authors’ response, which has largely addressed my concerns. I have also carefully reviewed the comments from other reviewers. Although the overall feedback is somewhat negative, I believe that further exploring KD from an information-theoretic perspective is still worthwhile and should be encouraged. I have therefore raised my score.

---

### Official Review · Reviewer_sE5L · 2025-11-01

**Soundness:** 2
**Presentation:** 2
**Contribution:** 2
**Rating:** 2
**Confidence:** 3

**Summary:**

This paper introduces the Cross-modal Complementarity Hypothesis (CCH), a simple criterion based on mutual information that enables users to a priori determine whether cross-modal knowledge distillation (KD) between teacher and student modalities is likely to be successful.
The work provides a theoretically motivated and practically meaningful attempt to define explicit conditions under which cross-modal KD is feasible.

**Strengths:**

- The paper presents an interesting and intuitive hypothesis based on comparisons of mutual information among the teacher, student, and label distributions.

 - The idea of formalizing a predictive criterion for when cross-modal KD will succeed is appealing and relevant to multimodal learning research.

- The experimental section is comprehensive and tried to empirically demonstrates the proposed hypothesis across multiple modalities.

**Weaknesses:**

- The theoretical analysis relies on simplified assumptions, including data modeled as a mixture of Gaussian distributions and the use of a linear regressor without non-linearity. These constraints substantially limit the generality of the theoretical findings. Since the theoretical proofs are derived under such restricted settings, the overall contributions are limited from a theoretical standpoint.
Moreover, the proposed criterion appears naïve and idealized, as it depends on assumptions such as access to optimal weights, which are impractical in real-world scenarios.

- Also, in the synthetic experiments, it remains unclear how the parameterization of the Gaussian mixture can be related to real-world datasets. Providing either a theoretical bridge or empirical demonstration of this connection would strengthen the paper.
Applying varying Gaussian blur to inputs intuitively disrupts cross-modal transfer, but it is not obvious how such manipulation validates Theorem 1. The link between this experimental setup and the theoretical claim should be clarified.

- Although the paper presents extensive empirical validation, the strength of evidence may not meet ICLR standards, given the simplicity of the theoretical setup and the lack of strong empirical grounding in realistic conditions.

**Questions:**

As noted above, there appears to be a disconnect between Theorem 1 and the experimental results. This point should be clarified, and I encourage the authors to provide further explanation or evidence to bridge this gap.

---

> ### Author Response · Authors · 2025-11-24
>
> **1. Weakness 1: The theoretical analysis relies on simplified assumptions, including data modeled as a mixture of Gaussian distributions and the use of a linear regressor without non-linearity. These constraints substantially limit the generality of the theoretical findings. Since the theoretical proofs are derived under such restricted settings, the overall contributions are limited from a theoretical standpoint. Moreover, the proposed criterion appears naïve and idealized, as it depends on assumptions such as access to optimal weights, which are impractical in real-world scenarios.**\
> **Weakness 3: Although the paper presents extensive empirical validation, the strength of evidence may not meet ICLR standards, given the simplicity of the theoretical setup and the lack of strong empirical grounding in realistic conditions.**
>
> Thank you for pointing this out. While our theoretical analysis indeed adopts simplified assumptions to enable a clean analytical characterization, **we have added a paragraph in Section 3** discussing how the criterion extends to non-linear theoretical settings. Moreover, our empirical results show that the proposed criterion remains accurate well beyond this idealized setting. In particular, Section 4 demonstrates strong agreement with theory across real-world datasets and nonlinear neural networks, confirming that the criterion is robust to non-Gaussianity and architectural complexity.
>
> Importantly, the criterion does not rely on access to optimal teacher weights. It only requires the teacher and student representations, which are directly observable. This makes the condition practical and easy to verify in realistic settings.
>
> **2. Weakness 2: Also, in the synthetic experiments, it remains unclear how the parameterization of the Gaussian mixture can be related to real-world datasets. Providing either a theoretical bridge or empirical demonstration of this connection would strengthen the paper. Applying varying Gaussian blur to inputs intuitively disrupts cross-modal transfer, but it is not obvious how such manipulation validates Theorem 1. The link between this experimental setup and the theoretical claim should be clarified.**\
> **Question 1: As noted above, there appears to be a disconnect between Theorem 1 and the experimental results. This point should be clarified, and I encourage the authors to provide further explanation or evidence to bridge this gap.**
>
> Thank you for raising this point. The Gaussian-mixture assumption is a standard abstraction used in theoretical analysis to derive clean information-theoretic expressions. To address concerns about real-data applicability, Section 4 empirically demonstrates that our criterion holds not only for Gaussian mixtures but also for real image, video, text and cancer datasets, which inherently deviate from Gaussian assumptions.
>
> Regarding the Gaussian-blur experiments: the purpose of applying blur is to systematically degrade the teacher’s representation quality in a controlled and quantifiable manner. Since our criterion depends only on the teacher representation $H_1$ and the student representation $H_2$, introducing blur decreases
> $I(H_1,H_2)$ while leaving $I(H_2,Y)$ unchanged. Our theory predicts that knowledge distillation becomes ineffective precisely when $I(H_1,H_2)$ drops below $I(H_2,Y)$. This transition is exactly what we observe in Section 4. Thus, the Gaussian-blur setup offers a practical and interpretable way to validate the theoretical prediction in a non-Gaussian, real-data regime.

---

### Official Review · Reviewer_qxvB · 2025-11-01

**Soundness:** 3
**Presentation:** 3
**Contribution:** 3
**Rating:** 6
**Confidence:** 2

**Summary:**

This paper proposes the Cross-modal Complementarity Hypothesis (CCH), an information-theoretic criterion for when cross-modal knowledge distillation (KD) helps. In a joint Gaussian setting, the authors show that KD improves a student when the mutual information between teacher and student representations exceeds that between the student and the label. Experiments across synthetic, vision, language, audio/video, and omics datasets, using multiple MI estimators, consistently align with this prediction, with gains when the MI gap is positive and degradations otherwise. The study provides a clear rationale for cross-modal KD and practical guidance on selecting teacher modalities and tuning KD strength.

**Strengths:**

1. The CCH gives a clear MI gap condition that is intuitive and can be checked with learned representations, turning “should we distill across modalities?” into a verifiable precondition and a practical handle for choosing the teacher modality and KD strength.

2. The paper provides a relatively rich theoretical analysis, which helps to make the imposed conditions more interpretable.

3. The experimental results on both synthetic and real-world tasks further validate the effectiveness of the proposed method.

**Weaknesses:**

1. The theoretical results are established under the assumption of joint Gaussian distributions. However, it remains unclear whether the sufficiency conditions still hold in non-Gaussian, nonlinear, and multi-class settings, as well as in the presence of practical asymmetries (e.g., heterogeneous architectures, representation mismatch, or domain-adaptive preprocessing). Further validation in these more realistic scenarios would strengthen the contribution.

2. When the student model has insufficient capacity, when excessive regularization is applied, or when the teacher model is overfitted, the behavior of the mutual-information-gap criterion appears to be insufficiently clarified.

3. When there are substantial differences between the teacher and student in terms of architecture, capacity, representation level, or domain preprocessing, it remains unclear whether the proposed method can still maintain reliable predictive performance.

**Questions:**

Does the proposed method remain effective when there are significant differences between the teacher and student architectures?

---

> ### Author Response · Authors · 2025-11-24
>
> **1. Weaknesses 1: The theoretical results are established under the assumption of joint Gaussian distributions. However, it remains unclear whether the sufficiency conditions still hold in non-Gaussian, nonlinear, and multi-class settings, as well as in the presence of practical asymmetries (e.g., heterogeneous architectures, representation mismatch, or domain-adaptive preprocessing). Further validation in these more realistic scenarios would strengthen the contribution.**
>
> Thank you for raising this point. While our theoretical analysis indeed relies on joint Gaussian assumptions, **we have added a paragraph in Section 3** discussing how the criterion extends to non-linear theoretical settings. We have also conducted extensive empirical studies demonstrating that our criterion remains accurate far beyond this idealized setting. In Section 4.1, we show that the criterion continues to hold under nonlinear models. Moreover, Sections 4.2–4.4 validate the criterion on real-world image, video, text, and cancer datasets, which naturally involve non Gaussianity, complex nonlinearities, and various practical asymmetries. These results suggest that the sufficiency conditions are robust in substantially more realistic scenarios than those covered by the theory.
>
> **2. Weakness 2: When the student model has insufficient capacity, when excessive regularization is applied, or when the teacher model is overfitted, the behavior of the mutual-information-gap criterion appears to be insufficiently clarified.**
>
> Thank you for your question. Since $H_1$ is obtained from the teacher and $H_2$ from the student, if the student has insufficient capacity or the teacher is overfitted, their representations will typically not satisfy our criterion $I(H_1; H_2) > I(H_2; Y)$. In such regimes, we indeed observe that KD does not help, which is consistent with the fact that our condition is sufficient but not necessary for successful KD. Regarding regularization, as discussed in Appendix A and in Figure 4 of Appendix B, when regularization is too strong, the benefit of KD diminishes and it becomes inefficient in practice.
>
> **3. Weakness 3: When there are substantial differences between the teacher and student in terms of architecture, capacity, representation level, or domain preprocessing, it remains unclear whether the proposed method can still maintain reliable predictive performance.**\
> **Question: Does the proposed method remain effective when there are significant differences between the teacher and student architectures?**
>
> Thank you for raising this concern. Our choice of using identical architectures for the teacher and student networks was intentional, as it allowed us to isolate the effect of mutual information without conflating it with capacity differences. To directly address the more realistic setting where the teacher has higher capacity than the student, **we have now added an additional synthetic experiment** in which the teacher network is strictly larger than the student, reported **in Appendix B**.

---

> > ### Comment · Reviewer_qxvB · 2025-11-26
> >
> > I appreciate the authors’ response. This paper relies on simplified assumptions, as also noted in the comments from other reviewers. However, my remaining concerns have been addressed, and I am therefore willing to raise my score to 6.

---

### Comment · Area_Chair_qLoz · 2025-11-24

Dear Reviewers,

Despite there being no rebuttals from the authors, **we still kindly encourage you to read other reviewers' comments and revise your ratings, if needed**. Your timely feedback is important for ensuring a fair and thorough review process. Thank you for your contributions to ICLR 2026.

AC

---

### Author Response · Authors · 2025-11-26

Dear Reviewers,

Thank you for your constructive feedback on our paper. Following our previous responses, we would like to kindly ask whether our clarifications have addressed the issues you raised. We remain available to answer any additional questions and are happy to provide further clarifications or revisions if needed.

We would also like to express our sincere gratitude to Reviewer k5cd for their encouragement and recognition of our work’s value. We strongly agree that exploring Knowledge Distillation (KD) from an information-theoretic perspective is a worthwhile endeavor, particularly for handling large, complex, and correlated datasets such as those found in healthcare.

To illustrate the practical utility of our method, consider the cancer data analysis in Section 4.4. The TCGA landscape includes a vast array of modalities—ranging from Electronic Health Records (EHRs) to genomics, proteomics, DNA methylation, and other omics data. In real-world scenarios, some modalities (such as single-cell RNA sequencing) are highly expensive to acquire, and we would like to further improve their performance. In such cases, given that estimating the mutual information takes only a few seconds, our Cross-modal Complementarity Hypothesis (CCH) offers a significant advantage. This allows practitioners to efficiently select the optimal “teacher” from a messy pool of candidates to boost the performance of weaker “student” modalities, avoiding the computational and time costs of exhaustive experimentation.

We hope this practical perspective highlights the contribution of our work, and we respectfully ask all reviewers to reconsider the value of our paper.

Thank you again for your valuable input.

Best regards,
The Authors

---

### Author Response · Authors · 2025-12-01

Dear Area Chair,

Thank you for handling our submission.

Our paper introduces the Cross-modal Complementarity Hypothesis (CCH), which provides a quantitative, testable information-theoretic criterion for when cross-modal knowledge distillation is beneficial. Rather than proposing yet another KD loss, we offer a decision rule: KD is predicted to help precisely when the mutual information between teacher and student representations exceeds that between the student representation and the labels. We prove sufficiency of this condition in a joint Gaussian setting and then show that the same MI-based criterion reliably predicts both gains and failures of KD across nonlinear neural networks and diverse real-world multimodal datasets (image, text, audio/video, and cancer omics).

In the rebuttal, we addressed the main concerns raised by the reviewers:

Scope of the theory: We clarified that the Gaussian analysis is intended as a minimal model that yields an exact and interpretable risk decomposition, and we added discussion and experiments showing that the CCH behavior persists well beyond this setting (teacher–student capacity gaps, multimodal learning fusion ).

MI estimation and practicality: We emphasized that CCH only needs the ordering of two MI terms (the “MI gap”), not precise values. Using representation-level estimators, we showed that this ordering is robust across different estimators and cheap to obtain compared to model training, making CCH a practical pre-distillation tool rather than a purely theoretical construct.

Relationship to prior intuitions (e.g., MFH and MI-maximizing KD): We clarified that, while prior work informally links KD success to shared label-relevant information, our contribution is to turn this into a concrete, computable inequality that (i) predicts when KD helps, (ii) warns when it will likely hurt, and (iii) guides teacher-modality selection. Our experiments illustrate this: a single MI check allows practitioners to choose among many expensive candidate teacher modalities without exhaustive KD trials.

We are encouraged that two reviewers explicitly raised their scores after considering these clarifications. We hope this summary helps convey both the conceptual clarity and the practical utility of CCH as a simple, general criterion for designing cross-modal KD in multimodal systems.

Thank you very much for your time and consideration.

Best regards,
Authors

---

### Meta-Review · Area_Chair_MnKZ · 2026-01-07

**Summary:**

This paper proposes the Cross-modal Complementarity Hypothesis (CCH), an information-theoretic criterion stating that cross-modal knowledge distillation (KD) is beneficial when the mutual information between teacher and student representations exceeds that between the student representation and the labels. The authors provide a sufficiency proof under a joint Gaussian and linear regression setting, and empirically validate the hypothesis across synthetic data and multiple multimodal benchmarks.

Reviewers raised substantial concerns regarding `the limited theoretical generality and practical applicability of the proposed framework`. In particular, the theoretical analysis relies on `highly idealized assumptions (joint Gaussian data, linear models, asymptotic regimes)` that significantly restrict its relevance to modern nonlinear deep learning settings. While the empirical results are extensive, several reviewers questioned whether they provide sufficiently strong or realistic evidence to compensate for the narrow theory, noting `a weak connection between the formal results and practical KD scenarios with heterogeneous architectures, capacity gaps, and finite-sample effects`. Additional concerns were raised about `the reliance on mutual information estimation`, which is known to be unstable and computationally challenging in high-dimensional settings, potentially undermining the claim that CCH serves as a truly a priori decision criterion.

Overall, most reviewers initially gave negative ratings of the paper, and the AC think the rebuttal would not significantly address the concerns on the work to meet the bar for a strong contribution. The combination of overly restrictive theoretical assumptions, questionable robustness of the MI-based criterion, and insufficiently convincing empirical grounding in realistic settings ultimately motivates a rejection decision.

**Reviewer Concerns:**

**1. [Partially addressed] Highly idealized theoretical assumptions and limited generality of the CCH framework**
(by: qxvB, sE5L, k5cd, akzG, UHPd)

Most of the reviewers raised concerns that the formal theoretical guarantees rely on strongly idealized assumptions, including joint Gaussian data distributions, linear regression models, and small-regularization or asymptotic regimes. They questioned whether these assumptions meaningfully reflect modern cross-modal knowledge distillation settings involving deep, nonlinear networks, heterogeneous architectures, and finite-sample training.

The authors attempted to address this concern by (i) adding a discussion of a non-linear setting in Section 4 and (ii) providing extensive empirical evidence across real-world datasets. However, from the AC's assessment, the non-linear discussion does not constitute a rigorous theoretical extension: while the loss reformulation is algebraically valid, the key mutual-information inequality underlying CCH in the non-linear case is assumed rather than derived, and the argument relies on heuristic reasoning (including an insufficient use of the data processing inequality). As such, this discussion does not fundamentally alter the idealized nature of the theory. Overall, the concern is only partially addressed.

**2. [Partially addressed] Limited theoretical novelty relative to prior intuitions and frameworks**
(by: `UHPd`, `nSot`)

Several reviewers argued that the central CCH criterion essentially formalizes a well-known intuition: cross-modal KD is beneficial when the teacher modality provides additional label-relevant information that the student lacks. From this perspective, the contribution was viewed as incremental, reframing existing ideas (e.g., MFH, MI-based KD) in information-theoretic terms rather than introducing a fundamentally new principle.

The authors clarified that their contribution lies in turning this intuition into a quantitative, computable inequality that can predict both gains and failures of KD, and supported this claim with empirical validation. While this strengthens the positioning, it remains unconvinced that this constitutes a substantial theoretical advance, leaving the novelty concern partially addressed.

**3. [Partially addressed] Practical reliability and robustness of mutual information estimation**
(by: k5cd, akzG, UHPd)

Reviewers highlighted that estimating mutual information in high-dimensional learned representations is notoriously difficult, estimator-dependent, and potentially unstable, calling into question the practicality of using CCH as an a priori decision rule. Concerns were raised about estimator variance, computational cost, and sensitivity to finite-sample noise.

The authors addressed this by emphasizing that CCH depends only on the relative ordering of two MI terms rather than precise estimates, and by showing consistency across multiple estimators and datasets. While this alleviates some concerns, it remains cautious about robustness at larger scales and in more complex settings, so the concern is considered partially addressed.

**4. [Partially addressed] Limited realism and scope of empirical validation**
(by: sE5L, nSot, UHPd)

Reviewers questioned whether the experimental setups sufficiently reflect realistic KD scenarios, noting controlled choices such as identical teacher–student architectures, relatively forgiving benchmarks (e.g., MNIST/MNIST-M), and limited stress-testing on harder vision-centric tasks or settings with large capacity gaps and noisy teachers.

The authors justified these choices as necessary to isolate mutual-information effects and added additional synthetic experiments with capacity differences. While reasonable, these explanations do not fully resolve concerns about external validity and practical deployment, leaving this issue partially addressed.

**Reviewer Scores:**

**Reviewer qxvB (already raised to 6)**

Reviewer qxvB appreciated the clarity of the theoretical formulation but raised concerns about the restrictive assumptions and limited generality to non-linear, practical KD settings. While the empirical validation helps, these core concerns remain only partially addressed. The reviewer have already increased the rating to 6 during discussion, so the score would likely remain unchanged.

**Reviewer sE5L (2 -> 2)**

Reviewer sE5L questioned the realism of the theoretical assumptions and the strength of empirical evidence. The additional discussion and experiments partially mitigate these concerns but do not fundamentally resolve them, suggesting no score change.

**Reviewer k5cd (already raised to 6)**

Reviewer k5cd focused on the difficulty and reliability of mutual information estimation in high-dimensional settings. Some of the concerns were addressed, and the reviewer raised the rating to 6 accordingly. From the AC perspective, there may not likely to have further rating increment.

**Reviewer akzG (4 -> 4)**

Reviewer akzG raised concerns including the generality of the CCH framework, the lack of a rigorous non-linear theoretical extension, the sensitivity of the experiments. While the paper provides useful intuition and empirical support, these issues remain partially unresolved, and the score would likely remain unchanged.

**Reviewer nSot (4 -> 4)**

Reviewer nSot viewed the contribution as conceptually appealing but incremental, and questioned whether the empirical evidence sufficiently supports the theoretical claims. Given that these concerns persist, no upward revision is expected.

**Reviewer UHPd (2 -> 2)**

Reviewer UHPd questioned the depth of theoretical novelty, the reliance on idealized assumptions, and the practicality of MI-based decision criteria. The paper’s responses partially address these points but do not change the overall evaluation, so the score would likely remain unchanged.

---

### Decision · Program_Chairs · 2026-01-26

Reject